# A differentiable Gillespie algorithm for simulating chemical kinetics, parameter estimation, and designing synthetic biological circuits

**Krishna Rijal\*, Pankaj Mehta**

Department of Physics, Boston University, Boston, United States

## eLife Assessment

This **important** study introduces a fully differentiable variant of the Gillespie algorithm as an approximate stochastic simulation scheme for complex chemical reaction networks, allowing kinetic parameters to be inferred from empirical measurements of network outputs using gradient descent. The concept and algorithm design are **convincing** and innovative. While the proofs of concept are promising, some questions are left open about implications for more complex systems that cannot be addressed by existing methods. This work has the potential to be of significant interest to a broad audience of quantitative and synthetic biologists.

**\*For correspondence:**
krishnarijal331@gmail.com

**Competing interest:** The authors declare that no competing interests exist.

**Abstract** The Gillespie algorithm is commonly used to simulate and analyze complex chemical reaction networks. Here, we leverage recent breakthroughs in deep learning to develop a fully differentiable variant of the Gillespie algorithm. The differentiable Gillespie algorithm (DGA) approximates discontinuous operations in the exact Gillespie algorithm using smooth functions, allowing for the calculation of gradients using backpropagation. The DGA can be used to quickly and accurately learn kinetic parameters using gradient descent and design biochemical networks with desired properties. As an illustration, we apply the DGA to study stochastic models of gene promoters. We show that the DGA can be used to: (1) successfully learn kinetic parameters from experimental measurements of mRNA expression levels from two distinct *Escherichia coli* promoters and (2) design nonequilibrium promoter architectures with desired input–output relationships. These examples illustrate the utility of the DGA for analyzing stochastic chemical kinetics, including a wide variety of problems of interest to synthetic and systems biology.

## Introduction

Randomness is a defining feature of our world. Stock market fluctuations, the movement of particles in fluids, and even the change of allele frequencies in organismal populations can all be described using the language of stochastic processes. For this reason, disciplines as diverse as physics, biology, ecology, evolution, finance, and engineering have all developed tools to mathematically model stochastic processes (*Van Kampen, 1992*; *Gardiner, 2009*; *Rolski et al., 2009*; *Wong and Hajek, 2012*). In the context of biology, an especially fruitful area of research has been the study of stochastic gene expression in single cells (*McAdams and Arkin, 1997*; *Elowitz et al., 2002*; *Raj and van Oudenaarden, 2008*; *Sanchez and Golding, 2013*). The small number of molecules involved in gene expression make stochasticity an inherent feature of protein production and numerous mathematical

and computational techniques have been developed to model gene expression and relate mathematical models to experimental observations (*Paulsson, 2005*; *Wilkinson, 2018*).

One prominent computational algorithm for understanding stochasticity in gene expression is the Gillespie algorithm, with its Direct Stochastic Simulation Algorithm variant being the most commonly used method (*Doob, 1945*; *Gillespie, 1977*). The Gillespie algorithm is an extremely efficient computational technique used to simulate the time evolution of a system in which events occur randomly and discretely (*Gillespie, 1977*). Beyond gene expression, the Gillespie algorithm is widely employed across numerous disciplines to model stochastic systems characterized by discrete, randomly occurring events including epidemiology (*Pineda-Krch, 2008*), ecology (*Parker and Kamenev, 2009*; *Dobramysl et al., 2018*), neuroscience (*Benayoun et al., 2010*; *Rijal et al., 2024*), and chemical kinetics (*Gillespie, 1976*; *Gillespie, 2007*).

Here, we revisit the Gillespie algorithm in light of the recent progress in deep learning and differentiable programming by presenting a 'fully differentiable' variant of the Gillespie algorithm we dub the differentiable Gillespie algorithm (DGA). The DGA modifies the traditional Gillespie algorithm to take advantage of powerful automatic differentiation libraries for example, PyTorch (*Paszke et al., 2019*), Jax (*Bradbury et al., 2018*), and Julia (*Bezanson et al., 2017*) and gradient-based optimization. The DGA allows us to quickly fit kinetic parameters to data and design discrete stochastic systems with a desired behavior. Our work is similar in spirit to other recent work that seeks to harness the power of differentiable programming to accelerate scientific simulations (*Liao et al., 2019*; *Schoenholz and Cubuk, 2020*; *Wei et al., 2019*; *Degrave et al., 2019*; *Arya et al., 2022*; *Arya et al., 2023*; *Bezgin et al., 2023*). The DGA's use of differential programming tools also complements more specialized numerical methods designed for performing parameter sensitivity analysis on Gillespie simulations such as finite-difference methods (*Anderson, 2012*; *Srivastava et al., 2013*; *Thanh et al., 2018*), the likelihood ratio method (*Glynn, 1990*; *McGill et al., 2012*; *Núñez and Vlachos, 2015*), and pathwise derivative methods (*Sheppard et al., 2012*).

One of the difficulties in formulating a differentiable version of the Gillespie algorithm is that the stochastic systems it treats are inherently discrete. For this reason, there is no obvious way to take derivatives with respect to kinetic parameters without making approximations. As shown in *Figure 1*, in the traditional Gillespie algorithm both the selection of the index for the next reaction and the updates of chemical species are both discontinuous functions of the kinetic parameters. To circumnavigate these difficulties, the DGA modifies the traditional Gillespie algorithm by approximating discrete operations with continuous, differentiable functions, smoothing out abrupt transitions to facilitate gradient computation via automatic differentiation (*Figure 1*). This significant modification preserves the core characteristics of the original algorithm while enabling integration with modern deep learning techniques.

One natural setting for exploring the efficacy of the DGA is recent experimental and theoretical works exploring stochastic gene expression. Here, we focus on a set of beautiful experiments that explore the effect of promoter architecture on steady-state gene expression (*Jones et al., 2014*). An especially appealing aspect of *Jones et al., 2014* is that the authors independently measured the kinetic parameters for these promoter architectures using orthogonal experiments. This allows us to directly compare the predictions of DGA to ground truth measurements of kinetic parameters. We then extend our considerations to more complex promoter architectures (*Lammers et al., 2023*) and illustrate how the DGA can be used to design circuits with a desired input–output relation.

## Results
### A differentiable approximation to the Gillespie algorithm

Before proceeding to discussing the DGA, we start by briefly reviewing how the traditional Gillespie algorithm simulates discrete stochastic processes. For concreteness, in our exposition, we focus on the chemical system shown in *Figure 1* consisting of three species, A, B, and C, whose abundances are described by a state vector $\mathbf{x} = (x_1, x_2, x_3)$. These chemical species can undergo $N = 3$ chemical reactions, characterized by rate constants, $r_i(\mathbf{x})$ where $i = 1, \ldots, 3$, and a stoichiometric matrix $S_{i\alpha}$ whose $i$th row encodes how the abundance $x_\alpha$ of species $\alpha$ changes due to reaction $i$. Note that in what follows, we will often supress the dependence of the rates $r_i(\mathbf{x})$ on $\mathbf{x}$ and simply write $r_i$.

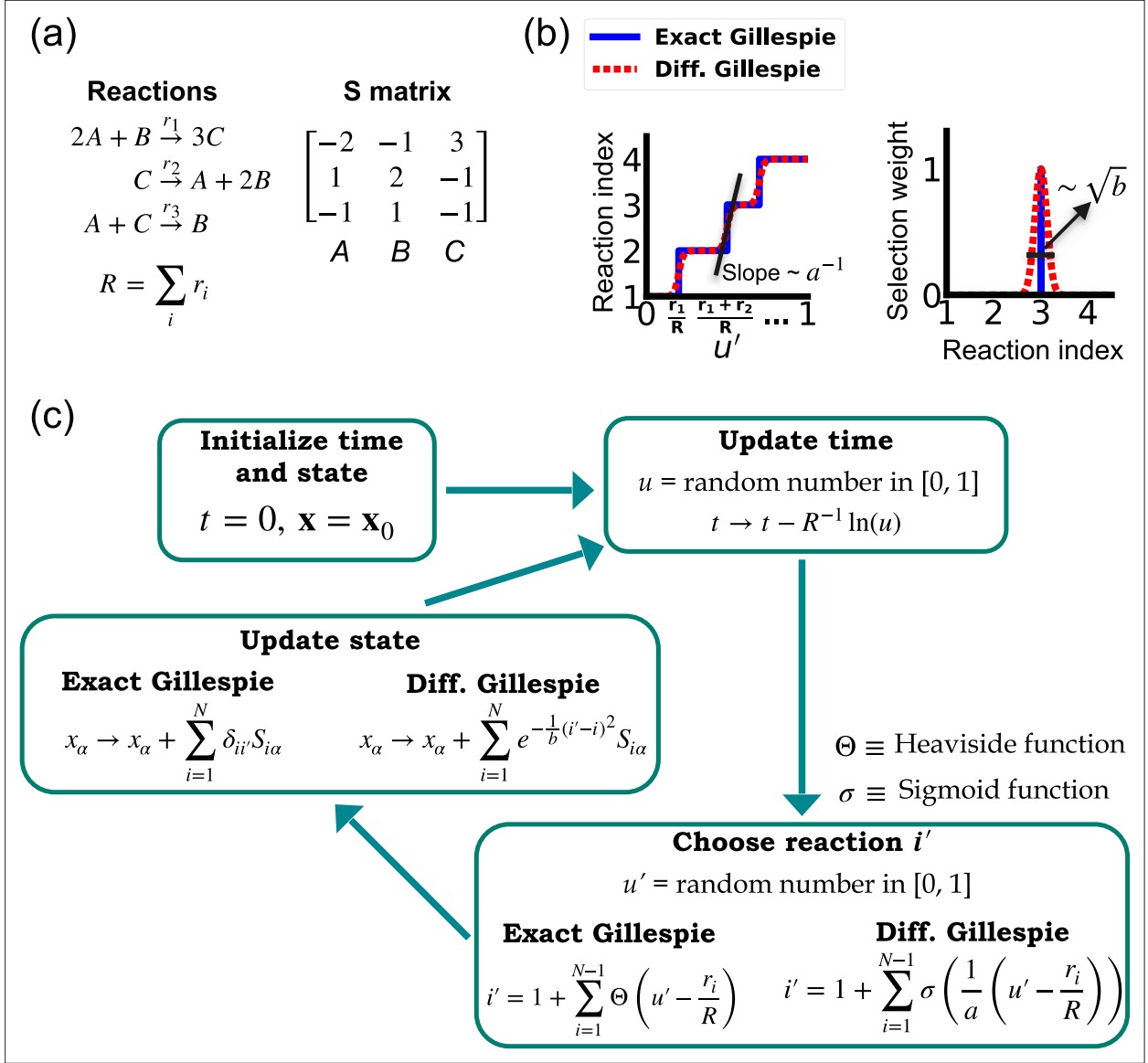

**Figure 1.** Comparison between the exact Gillespie algorithm and the differentiable Gillespie algorithm (DGA) for simulating chemical kinetics. (**a**) Example of kinetics with $N = 3$ reactions with rates $r_i(i = 1, 2, 3)$. (**b**) Illustration of the DGA's approximations: replacing the non-differentiable Heaviside and Kronecker delta functions with smooth sigmoid and Gaussian functions, respectively. (**c**) Flow chart comparing exact and differentiable Gillespie simulations.

In order to simulate such a system, it is helpful to discretize time into small intervals of size $\Delta t \ll 1$. The probability that a reaction $i$ with rate $r_i$ occurs during such an interval is simply $r_i \Delta t$. By construction, we choose $\Delta t$ to be small enough that $r_i \Delta t \ll 1$ and that the probability that a reaction occurs in any interval $\Delta t$ is extremely small and well described by a Poisson process. This means that naively simulating such a process is extremely inefficient because, in most intervals, no reactions will occur.

### Gillespie algorithm

The Gillespie algorithm circumnavigates this problem by: (1) exploiting the fact that the reactions are independent so that the rate at which *any* reaction occurs is also described by an independent Poisson process with rate $R = \Sigma_i r_i$ and (2) the waiting time distribution $p(\tau)$ of a Poisson process with rate $R$ is the exponential distribution $p(\tau) = Re^{-R\tau}$. The basic steps of the Gillespie algorithm are illustrated in *Figure 1*.

The simulation begins with the initialization of time and state variables:

$$t = 0,$$

$$\mathbf{x} = \mathbf{x}_0,$$

where $t$ is the simulation time. One then samples the waiting time distribution $p(\tau)$ for a reaction to occur to determine when the next the reaction occurs. This is done by drawing a random number $u$ from a uniform distribution over $[0, 1]$ and updating

$$t \to t - R^{-1} \ln(u). \tag{1}$$

Note that this time update is a fully differentiable function of the rates $r_i$.

In order to determine which of the reactions $i'$ occurs after a time $\tau$, we note that probability that reaction $i$ occurs is simply given by $q_i = r_i/R$. Thus, we can simply draw another random number $u'$ and choose $i'$ such that $i'$ equals the smallest integer satisfying

$$\sum_{i=1}^{i'} r_i/R > u'. \tag{2}$$

The reaction abundances $\mathbf{x}$ are then updated using the stoichiometric matrix

$$x_\alpha \to x_\alpha + S_{i'\alpha}. \tag{3}$$

Unlike the time update, both the choice of the reaction $i'$ and the abundance updates are not differentiable since the choice of the reaction $i'$ is a discontinuous function of the parameters $r_i$.

## Approximating updates in the Gillespie with differentiable functions

In order to make use of modern deep learning techniques and modern automatic differentiation packages, it is necessary to modify the Gillespie algorithm in such as way as to make the choice of reaction index (*Equation 2*) and abundance updates (*Equation 3*) differentiable functions of the kinetic parameters. To do so, we rewrite *Equation 2* using a sum of Heaviside step function $\Theta(y)$ (recall $\Theta(y) = 0$ if $y < 0$ and $\Theta(y) = 1$ if $y > 0$):

$$i' = 1 + \sum_{i=1}^{N-1} \Theta\left(u' - \frac{r_i}{R}\right). \tag{4}$$

This formulation of index selection makes clear the source of non-differentiability. The derivative of the $i'$ with respect to $r_i$ does not exist at the transition points where the Heaviside function jumps (see *Figure 1b*).

This suggests a natural modification of the Gillespie algorithm to make it differentiable – replacing the Heaviside function $\Theta(y)$ by a sigmoid function of the form

$$\sigma(y) = \frac{1}{1 + e^{-\frac{y}{a}}}, \tag{5}$$

where we have introduced a 'hyper-parameter' $a$ that controls the steepness of the sigmoid and plays an analogous role to temperature in a Fermi function in statistical mechanics. A larger value of $a^{-1}$ results in a steeper slope for the sigmoid functions, thereby more closely approximating the true Heaviside functions which is recovered in the limit $a \to 0$ (see *Figure 1b*). With this replacement, the index selection equation becomes

$$i' = 1 + \sum_{i=1}^{N-1} \sigma\left(\frac{1}{a}\left(u' - \frac{r_i}{R}\right)\right) \tag{6}$$

Note that in making this approximation, our index is no longer an integer, but instead can take on all real values between 0 and $N$. However, by making $a$ sufficiently small, *Equation 6* still serves as a good approximation to the discrete jumps in *Equation 4*. In general, $a$ is a hyperparameter that is chosen to be as small as possible while still ensuring that the gradient of $i'$ with respect to the kinetic

parameters $r_i$ can be calculated numerically with high accuracy. For a detailed discussion, please see *Appendix 1—figure 1* and Appendix 1.

Since the index $i'$ is no longer an integer but a real number, we must also modify the abundance update in *Equation 3* to make it fully differentiable. To do this, we start by rewriting *Equation 3* using the Kronecker delta $\delta_{ij}$ (where $\delta_{ij} = 1$ if $i = j$ and $\delta_{ij} = 0$ if $i \neq j$) as

$$x_\alpha \to x_\alpha + \sum_{i=1}^{N} \delta_{ii'} S_{i\alpha} \tag{7}$$

Since $i'$ is no longer an integer, we can approximate the Kronecker delta $\delta_{ii'}$ by a Gaussian function, to arrive at the approximate update equation

$$x_\alpha \to x_\alpha + \sum_{i=1}^{N} e^{-\frac{1}{b}(i'-i)^2} S_{i\alpha}. \tag{8}$$

The hyperparameter $b$ is generally chosen to be as small as possible while still ensuring numerical stability of gradients (*Appendix 1—figure 1*). Note by using an abundance update of the form *Equation 8*, the species abundances $\mathbf{x}$ are now real numbers. This is in stark contrast with the exact Gillespie algorithm where the abundance update (*Equation 7*) ensures that the $x_\alpha$ are all integers.

## Combining the DGA with gradient-based optimization

The goal of making Gillespie simulations differentiable is to enable the computation of the gradient of a *loss function*, $L(\boldsymbol{\theta})$, with respect to the kinetics parameters $\boldsymbol{\theta}$. A loss function quantifies the difference between simulated and desired values for quantities of interest. For example, when employing the DGA in the context of fitting noisy gene expression models, a natural choice for $L(\boldsymbol{\theta})$ is the difference between the simulated and experimentally measured moments of mRNA/protein expression (or alternatively, the Kullback–Leibler divergence between the experimental and simulated mRNA/protein

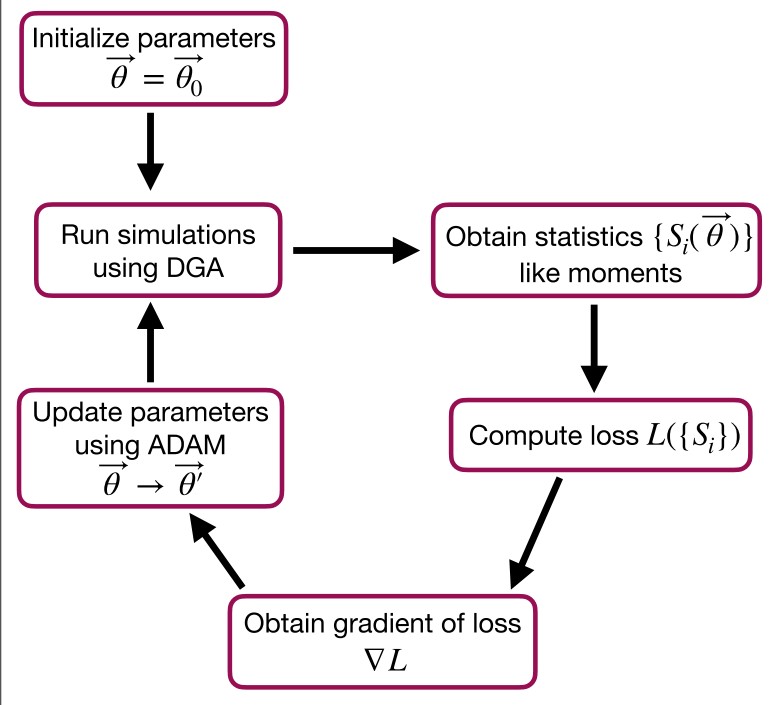

**Figure 2.** Flow chart of the parameter optimization process using the differentiable Gillespie algorithm (DGA). The process begins by initializing the parameters $\vec{\theta} = \vec{\theta_0}$. Simulations are then run using the DGA to obtain statistics $\{S_i(\vec{\theta})\}$ like moments. These statistics are used to compute the loss $L(\{S_i\})$, and the gradient of the loss $\nabla L$ is obtained. Finally, parameters are updated using the ADAM optimizer, and the process iterates to minimize the loss.

expression distributions if full distributions can be measured). When using the DGA to design gene circuits, the loss function can be any function that characterizes the difference between the simulated and desired values of the input–output relation.

The goal of the optimization using the DGA is to find parameters $\theta$ that minimize the loss. The basic workflow of a DGA-based optimization is shown in *Figure 2*. One starts with an initial guess for the parameters $\theta_0$. One then uses DGA algorithm to simulate the systems and calculate the gradient of the loss function $\nabla_\theta L(\theta)$. One then updates the parameters, moving in the direction of the gradient using gradient descent or more advanced methods such as ADAM (*Kingma and Ba, 2014*; *Mehta et al., 2019*), which uses adaptive estimates of the first and second moments of the gradients to speed up convergence to a local minimum of the loss function.

## The price of differentiability

A summary of the DGA is shown in *Figure 1*. Unsurprisingly, differentiability comes at a price. The foremost of these is that unlike the Gillespie algorithm, the DGA is no longer exact. The DGA replaces the exact discrete stochastic system by an approximate differentiable stochastic system. This is done by allowing both the reaction index and the species abundances to be continuous numbers. Though in theory, the errors introduced by these approximations can be made arbitrarily small by choosing the hyper-parameters $a$ and $b$ small enough (see *Figure 1*), in practice, gradients become numerically unstable when $a$ and $b$ are sufficiently small (see Appendix 1 and *Appendix 1—figure 1*).

In what follows, we focus almost exclusively on steady-state properties that probe the 'bulk', steady-state properties of the stochastic system of interest. We find the DGA works well in this setting. However, we note that the effect of the approximations introduced by the DGA may be pronounced in more complex settings such as the calculation of rare events, modeling of tail-driven processes, or dealing with non-stationary time series.

In order to better understand the DGA in the context of stochastic gene expression, we benchmarked the DGA on a simple two-state promoter model inspired by experiments in *Escherichia coli* (*Jones et al., 2014*). This simple model had several advantages that make it well suited for exploring the performance of DGA. These include the ability to analytically calculate mRNA expression distributions and independent experimental measurements of kinetic parameters.

## Two-state promoter model

Gene regulation is tightly regulated at the transcriptional level to ensure that genes are expressed at the right time, place, and in the right amount (*Phillips et al., 2012*). Transcriptional regulation involves various mechanisms, including the binding of transcription factors to specific DNA sequences, the modification of chromatin structure, and the influence of non-coding RNAs, which collectively control the initiation and rate of transcription (*Phillips et al., 2012*; *Sanchez et al., 2013*; *Phillips et al., 2019*).

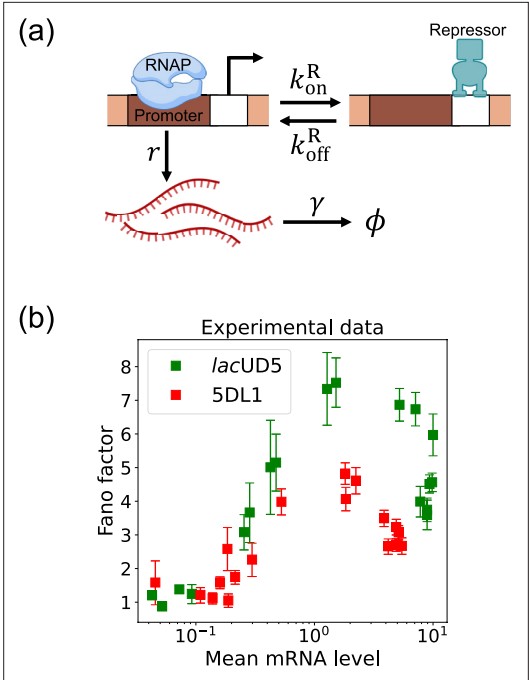

**Figure 3.** Two-state gene regulation architecture. (**a**) Schematic of gene regulatory circuit for transcriptional repression. RNA polymerase (RNAP) binds to the promoter region to initiate transcription at a rate $r$, leading to the synthesis of mRNA molecules (red curvy lines). mRNA is degraded at a rate $\gamma$. A repressor protein can bind to the operator site, with association and dissociation rates $k_{on}^R$ and $k_{off}^R$, respectively. (**b**) Experimental data from *Phillips et al., 2019*, showing the relationship between the mean mRNA level and the Fano factor for two different promoter constructs: *lac*UD5 (green squares) and 5DL1 (red squares).

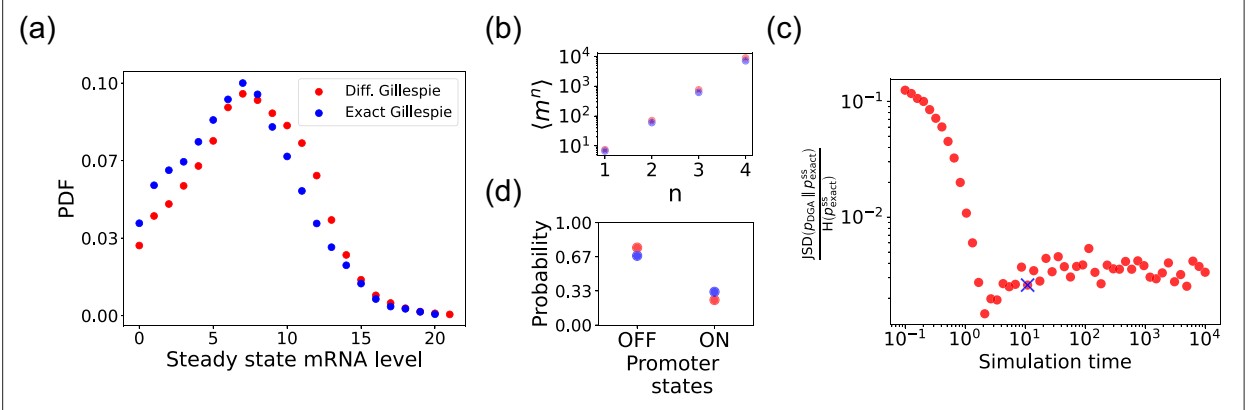

**Figure 4.** Accuracy of the differentiable Gillespie algorithm (DGA) in simulating the two-state promoter architecture in **Figure 3a**. Comparison between the DGA and exact simulations for (**a**) steady-state mRNA distribution, (**b**) moments of the steady-state mRNA distribution, and (**d**) the probability for the promoter to be in the 'ON' or 'OFF' state. (**c**) Ratio of the Jensen–Shannon divergence $\mathrm{JSD}\left(p_{\mathrm{DGA}} \| p_{\mathrm{exact}}^{\mathrm{ss}}\right)$ between the differentiable Gillespie probability distribution function (PDF) $p_{\mathrm{DGA}}$ and the exact steady-state PDF $p_{\mathrm{exact}}^{\mathrm{ss}}$, and the Shannon entropy $\mathrm{H}(p_{\mathrm{exact}}^{\mathrm{ss}})$ of the exact steady-state PDF. In all of the plots, 2000 trajectories are used. The simulation time used in panels (**a**), (**b**), and (**d**) is marked by blue 'x'. Parameter values: $k_{\mathrm{on}}^{\mathrm{R}} = 0.5$, $k_{\mathrm{off}}^{\mathrm{R}} = 1.0$, $r = 10$, $\gamma = 1$, $1/a = 200$, and $1/b = 20$.

By orchestrating these regulatory mechanisms, cells can respond to internal signals and external environmental changes, maintaining homeostasis and enabling proper development and function.

Here, we focus on a classic two-state promoter gene regulation (**Jones et al., 2014**). Two-state promoter systems are commonly studied because they provide a simplified yet powerful model for understanding gene regulation dynamics. These systems, characterized by promoters toggling between active and inactive states, offer insights into how genes are turned on or off in response to various stimuli (see **Figure 3a**). The two-state gene regulation circuit involves the promoter region, where RNA polymerase (RNAP) binds to initiate transcription and synthesize mRNA molecules at a rate $r$. A repressor protein can also bind to the operator site at a rate $k_{\mathrm{on}}^{\mathrm{R}}$ and unbind at a rate $k_{\mathrm{off}}^{\mathrm{R}}$. When the repressor is bound to the operator, it prevents RNAP from accessing the promoter, effectively turning off transcription. mRNA is also degraded at a rate $\gamma$. An appealing feature of this model is that both mean mRNA expression and the Fano factor can be calculated analytically and there exist beautiful quantitative measurements of both these quantities (**Figure 3b**). For this reason, we use this two-state promoters to benchmark the efficacy of DGA below.

## Characterizing errors due to approximations in the DGA

We begin by testing the DGA to do forward simulations on the two-state promoter system described above and comparing the results to simulations performed with the exact Gillespie algorithm (see Appendix 2 for simulation details). **Figure 4a** compares the probability distribution function (PDF) for the steady-state mRNA levels obtained from the DGA (in red) and the exact Gillespie simulation (in blue). The close overlap of these distributions demonstrates that the DGA can accurately replicate the results of the exact Gillespie simulation. This is also shown by the very close match of the first four moments $\langle m^n \rangle$ of the mRNA count between the exact Gillespie and the DGA in **Figure 4b**, though the DGA systematically overestimates these moments. As observed in **Figure 4a**, the DGA also fails to accurately capture the tails of the underlying PDF. This discrepancy arises because rare events result from very frequent low-probability reaction events where the sigmoid approximation used in the DGA significantly impacts the reaction selection process and, consequently, the final simulation results.

Next, we compare the accuracy of the DGA in simulating mRNA abundance distributions across a range of simulation times (see **Figure 4c**). The accuracy is quantified by the ratio of the Jensen–Shannon divergence $\mathrm{JSD}\left(p_{\mathrm{DGA}} \| p_{\mathrm{exact}}^{\mathrm{ss}}\right)$ between the differentiable Gillespie PDF $p_{\mathrm{DGA}}$ and the exact steady-state PDF $p_{\mathrm{exact}}^{\mathrm{ss}}$, and the entropy $\mathrm{H}(p_{\mathrm{exact}}^{\mathrm{ss}})$ of the exact steady-state PDF. For probability distributions $P$ and $Q$ over the same discrete space $\mathcal{X}$, the JSD and H are defined as:

$$\text{JSD}(P \parallel Q) = \frac{1}{2}D_{\text{KL}}(P \parallel M) + \frac{1}{2}D_{\text{KL}}(Q \parallel M)$$

$$\text{H}(P) = -\sum_{x \in \mathcal{X}} P(x) \log P(x)$$

(9)

where $M = \frac{1}{2}(P + Q)$ and $D_{\text{KL}}$ denotes the Kullback–Leibler divergence

$$D_{\text{KL}}(P \parallel Q) = \sum_{x \in \mathcal{X}} P(x) \log \frac{P(x)}{Q(x)}$$

(10)

The ratio $\frac{\text{JSD}}{\text{H}}$ normalizes divergence by entropy, enabling meaningful comparison across systems. As expected, the $\frac{\text{JSD}}{\text{H}}$ ratio decreases with increasing simulation time, indicating convergence toward the steady-state distribution of the exact Gillespie simulation. By 'steady-state distribution', we mean the long-term probability distribution of states that the exact Gillespie algorithm approaches after a simulation time of $10^4$. The saturation of the $\frac{\text{JSD}}{\text{H}}$ ratio at approximately 0.003 for long simulation times is due to the finite values of $a^{-1}$ and $b^{-1}$. In percentage terms, this ratio represents a 0.3% divergence, meaning that the DGA's approximation introduces only a 0.3% deviation from the exact distribution, relative to the total uncertainty (entropy) in the exact system.

Finally, the bar plot in *Figure 4d* shows simulation results for the probability of the promoter being in the 'OFF' and 'ON' states as predicted by the DGA (in red) and the exact Gillespie simulation (in blue). The differentiable Gillespie overestimates the probability of being in the 'OFF' state and underestimates the probability of being in the 'ON' state. Nonetheless, given the discrete nature of this system, the DGA does a reasonable job of matching the results of the exact simulations.

As we will see below, despite these errors the DGA is able to accurately capture gradient information and hence works remarkably well at gradient-based optimization of loss functions.

## Parameter estimation using the DGA

In many applications, one often wants to estimate kinetic parameters from experimental measurements of a stochastic system (*Tian et al., 2007*; *Munsky et al., 2009*; *Komorowski et al., 2009*; *Villaverde et al., 2019*). For example, in the context of gene expression, biologists are often interested in understanding biophysical parameters such as the rate at which promoters switch between states or a transcription factor unbinds from DNA. However, estimating kinetic parameters in stochastic systems poses numerous challenges because the vast majority of methods for parameter estimation are designed with deterministic systems in mind. Moreover, it is often difficult to analytically calculate likelihood functions making it difficult to perform statistical inference. One attractive method for addressing these difficulties is to combine differentiable Gillespie simulations with gradient-based optimization methods. By choosing kinetic parameters that minimize the difference between simulations and experiments as measured by a loss function, one can quickly and efficiently estimate kinetic parameters and error bars.

## Loss function for parameter estimation

To use the DGA for parameter estimation, we start by defining a loss function $L(\boldsymbol{\theta})$ that measures the discrepancy between simulations and experiments. In the context of the two-state promoter model (*Figure 3*), a natural choice of loss function is the square error between the simulated and experimentally measured mean and standard deviations of the steady-state mRNA distributions:

$$L(\boldsymbol{\theta}) = (\langle \hat{m} \rangle - \langle m \rangle)^2 + (\hat{\sigma}_m - \sigma_m)^2,$$

(11)

where $\langle \hat{m} \rangle$ and $\hat{\sigma}_m$ denote the mean and standard deviation obtained from DGA simulations, and $\langle m \rangle$ and $\sigma_m$ are the experimentally measured values of the same quantities. Having specified the loss function and parameters, we then use the gradient-based optimization to minimize the loss and find the optimal parameters $\hat{\boldsymbol{\theta}}$ (see *Figure 2*). Note that in general the solution to the optimization problem need not be unique (see below).

## Confidence intervals and visualizing loss landscapes

Given a set of learned parameters $\hat{\theta}$ that minimize $L(\theta)$, one would also ideally like to assign a confidence interval (CI) to this estimate that reflect how constrained these parameters are. One natural way to achieve this is by examining the curvature of the loss function as the parameter $\theta_i$ varies around its minimum value, $\theta_i^{\min}$. Motivated by this, we define the 95% CIs for parameter $\theta_i$ by:

$$CI_{\theta_i} = [\theta_i^{\min} - \delta, \theta_i^{\min} + 1.96\delta_{\theta_i}] \tag{12}$$

where

$$\delta_{\theta_i} = \left( \sqrt{\frac{\partial^2 L}{\partial \theta_i^2}} \right)^{-1} \Bigg|_{\theta_i = \theta_i^{\min}} \tag{13}$$

and $L(\theta_i^{\min} - \delta) = L(\theta_i^{\min} + 1.96\delta_{\theta_i})$. A detailed explanation of how to numerically estimate the CIs is given in Appendix 3.

One shortcoming of *Equation 13* is that it treats each parameter in isolation and ignores correlations between parameters. On a technical level, this is reflected in the observation that the CIs only know about the diagonal elements of the full Hessian $\partial_{ij}^2 L(\theta)$. This shortcoming is especially glaring when there are many sets of parameters that all optimize the loss function (*Einav et al., 2018*; *Razo-Mejia et al., 2018*). As discuss below, this is often the case in many stochastic systems including the two-state promoter architecture in *Figure 3*. For this reason, it is often useful to make two dimensional plots of the loss function $L(\theta)$. To do so, for each pair of parameters, we simply sample the parameters around their optimal value and forward simulate to calculate the loss function $L(\theta)$. We then use this simulations to create two-dimensional heat maps of the loss function. This allows us to identify 'soft directions' in parameter space, where the loss function $L(\theta)$ changes slowly, indicating weak sensitivity to specific parameter combinations.

## Parameter estimation on synthetic data

Before proceeding to experiments, we start by benchmarking the DGA's ability to perform parameter estimation on synthetic data generated using the two-state promoter model shown in *Figure 3*. This model nominally has four independent kinetic parameters: the rate at which repressors bind the promoter, $k_{on}^R$; the rate at which the repressor unbinds from the promoter, $k_{off}^R$; the rate at which mRNA is produced, $r$; and the rate at which mRNA degrades, $\gamma$. Since we are only concerned with steady-state properties of the mRNA distribution, we choose to measure time in units of the off rate and set $k_{off}^R = 1$ in everything that follows. In Appendix 4, we make use of exact analytical results for $\langle m \rangle$ and $\sigma_m$ to show that the solution to the optimization problem specified by loss function in *Equation 11* is degenerate – there are many combinations of the three parameters $\{k_{on}^R, r, \gamma\}$ that all optimize $L(\theta)$. On the other hand, if one fixes the mRNA degradation rate $\gamma$, this degeneracy is lifted and there is a unique solution to the optimization problem for the two parameters $\{k_{on}^R, r\}$. We discuss both these cases below.

## Generating synthetic data

To generate synthetic data, we randomly sample the three parameters: $k_{on}^R$, $r$, and $\gamma$ within the range $[0.1, 10]$, while keeping $k_{off}^R$ fixed at 1. In total, we generate 20 different sets of random parameters. We then perform exact Gillespie simulations for each set of parameters. Using these simulations, we obtain the mean $\langle m \rangle$ and standard deviation $\sigma_m$ of the mRNA levels, which are then used as input to the loss function in *Equation 11*. We then use the DGA to estimate the parameters using the procedure outlined above and compare the resulting predictions with ground truth values for simulations.

## Estimating parameters in the non-degenerate case

We begin by considering the case where the mRNA degradation rate $\gamma$ is known and the goal is to estimate the two other parameters: the repressor binding rate $k_{on}^R$ and the mRNA production rate $r$. As discussed above, in this case, the loss function in *Equation 11* has a unique minima, considerably simplifying the inference task. *Figure 5a* shows a scatter plot of the learned and the true parameter values for wide variety of choices of $\gamma$. As can be seen, there is very good agreement between

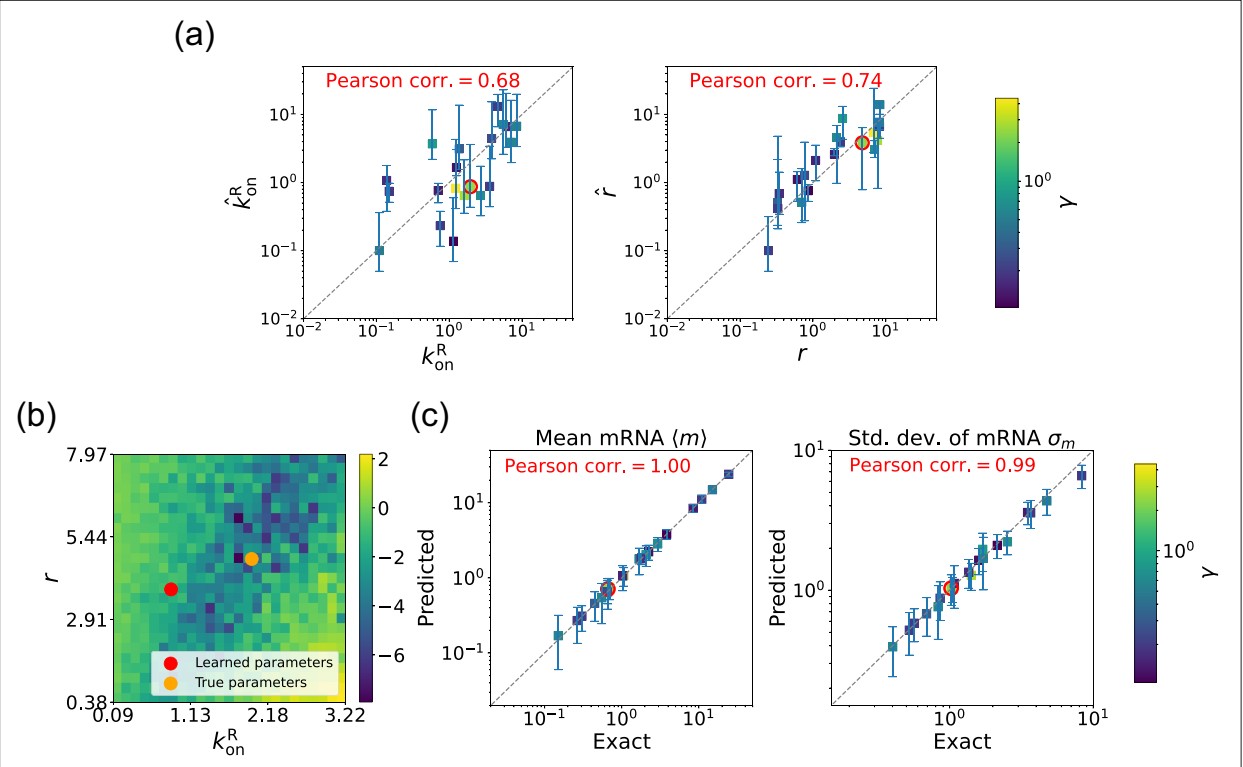

**Figure 5.** Gradient-based learning via differentiable Gillespie algorithm (DGA) is applied to the synthetic data for the gene expression model in *Figure 3a*. Parameters $k_{off}^R$ are fixed at 1, with $1/a = 200$ and $1/b = 20$ for a simulation time of 10. (a) Scatter plot of true versus inferred parameters ($\hat{k}_{on}^R$ and $\hat{r}$) with $\gamma$ constant. Error bars are 95% confidence intervals (CIs). Panel (b) plots the logarithm of the loss function near a learned parameter set (shown in red circles in (a)), showing insensitivity regions. Panel (c) compares true and predicted mRNA mean and standard deviation with 95% CIs.

the true parameters and learned parameters. *Figure 5c* shows that even when the true and learned parameters differ, the DGA can predict the mean $\langle m \rangle$ and standard deviation $\sigma_m$ of the steady-state mRNA distribution almost perfectly (see Appendix 5 for discussion of how error bars were estimated). To better understand this, we selected a set of learned parameters: $k_{on}^R = 0.87$, $r = 3.83$, and $\gamma = 2.43$. We then plotted the loss function in the neighborhood of these parameters (*Figure 5b*). As can be seen, the loss function around the true parameters is quite flat and the learned parameters live at the edge of this flat region. The flatness of the loss function reflects the fact that the mean and standard deviation of the mRNA distribution depend weakly on the kinetic parameters.

## Estimating parameters for the degenerate case

We now estimate parameters for the two-state promoter model when all three parameters $k_{on}^R$, $r$, and $\gamma$ are unknown. As discussed above, in this case, there are many sets of parameters that all minimize the loss function in *Equation 11*. *Figure 6a* shows a comparison between the learned and true parameters along with a heat map of the loss function for one set of synthetic parameters (*Figure 6b*). As can be seen in the plots, though the true parameters and learned parameter values differ significantly, they do so along 'sloppy' directions where loss function is flat. Consistent with this intuition, we performed simulations comparing the mean $\langle \hat{m} \rangle$ and standard deviation $\hat{\sigma}_m$ of the steady-state mRNA levels using the true and learned parameters and found near-perfect agreement across all of the synthetic data (*Figure 6c*).

## Parameter estimation on experimental data

In the previous section, we demonstrated that our DGA can effectively obtain parameters for synthetic data. However, real experimental data often contains noise and variability, which can complicate the parameter estimation process. To test the DGA in this more difficult setting, we reanalyze experiments by *Jones et al., 2014* which measured how mRNA expression changes in a system well described by

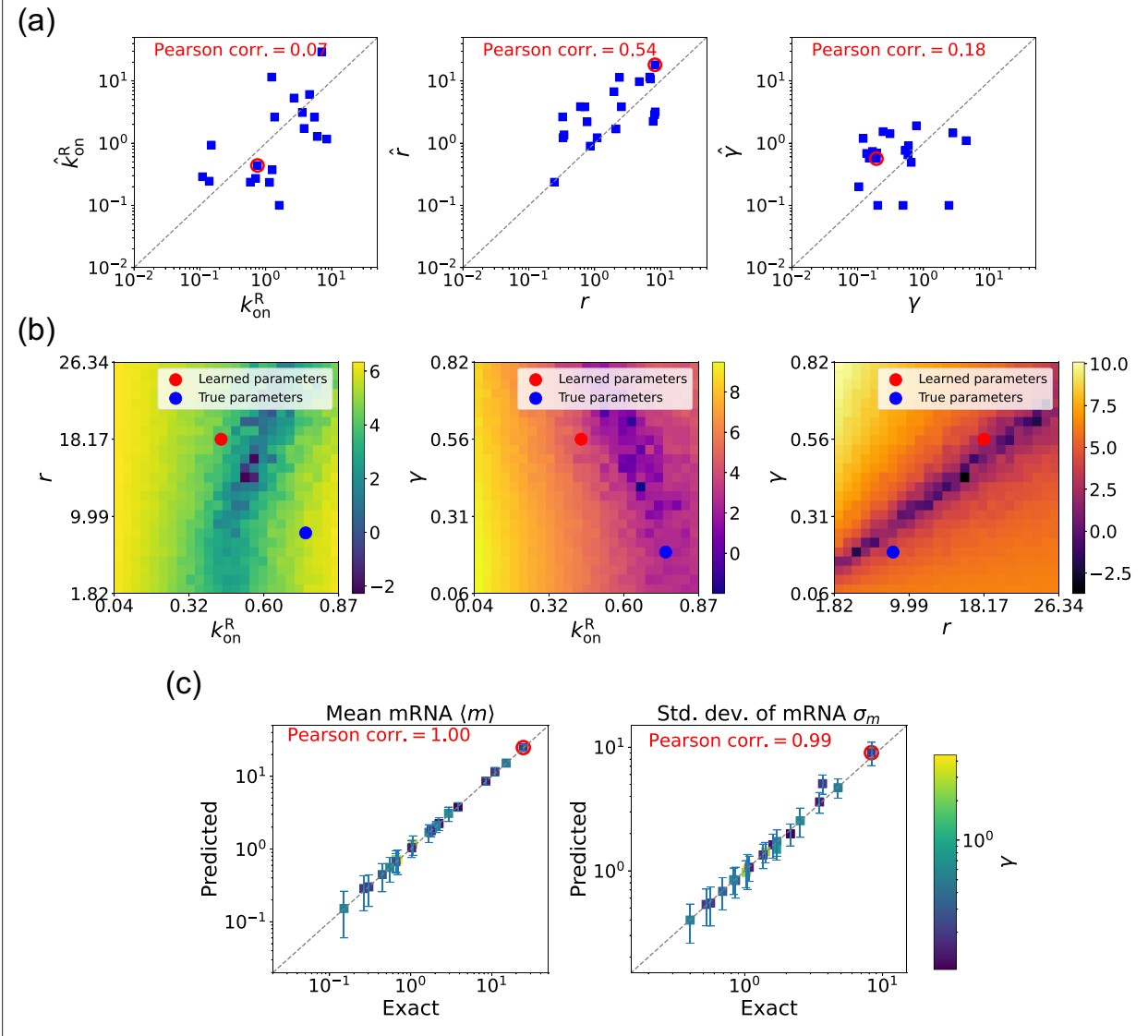

**Figure 6.** Gradient-based learning via differentiable Gillespie algorithm (DGA) is applied to the synthetic data for the gene expression model in **Figure 3a**. Parameters $k_{off}^R$ are fixed at 1, with $1/a = 200$ and $1/b = 20$ for a simulation time of 10. (**a**) Scatter plot of true versus inferred parameters ($\hat{k}_{on}^R$, $\hat{r}$, and $\gamma$). Error bars are 95% confidence intervals (CIs). Panel (**b**) plots the logarithm of the loss function near a learned parameter set (shown in red circles in (a)), showing insensitivity regions. Panel (**c**) compares true and predicted mRNA mean and standard deviation with 95% CIs.

the two-state gene expression model in **Figure 3**. In these experiments, two constitutive promoters *lac*UD5 and 5DL1 (with different transcription rates $r$) were placed under the control of a LacI repressor through the insertion of a LacI binding site. By systematically varying LacI concentrations, the authors were able to adjust the repressor binding rate $k_{on}^R$. mRNA fluorescence in situ hybridization was employed to measure mRNA expression, providing data on both mean expression levels $\langle m \rangle$ and the variability as quantified by the Fano factor $f = \sigma_m^2/\langle m \rangle$ for both promoters (see **Figure 3b**).

Given a set of measurements of the mean and Fano factor $\{\langle m \rangle_i, f_i^m\}$ for a promoter (*lac*UD5 and 5DL1), we construct a loss function of the form:

$$L = \sum_{i=1}^{N}(\langle \hat{m} \rangle_i - \langle m \rangle_i)^2 + \sum_{i=1}^{N}(\hat{\sigma}_i^m - \sqrt{f_i^m \langle m \rangle_i})^2, \quad (14)$$

where $i$ runs over data points (each with a different lac repressor concentration) and $\langle \hat{m} \rangle_i$ and $\hat{\sigma}_i^m$ are the mean and standard deviation obtained from a sample of DGA simulations. This loss function is

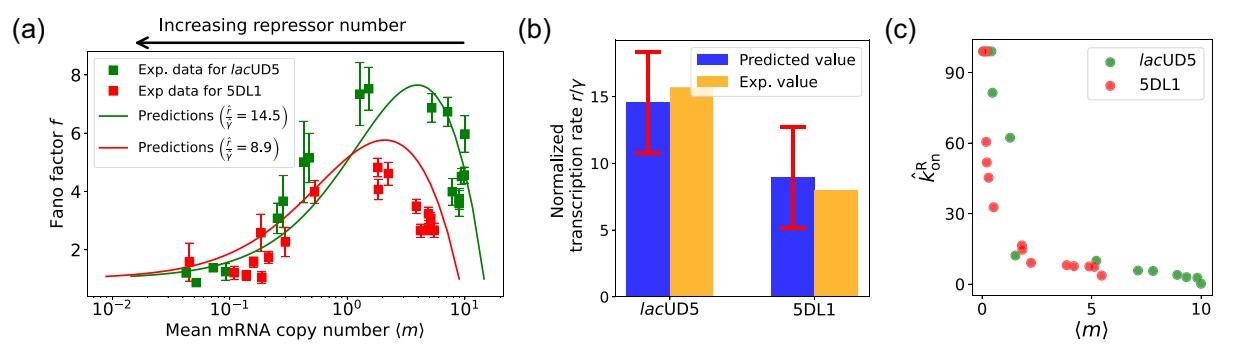

**Figure 7.** Fitting of experimental data from *Jones et al., 2014* using the differentiable Gillespie algorithm (DGA). (**a**) Comparison between theoretical predictions from the DGA (solid curves) and experimental values of mean and the Fano factor for the steady-state mRNA levels are represented by square markers, along with the error bars, for two different promoters, *lac*UD5 and 5DL1. Solid curves are generated by using DGA to estimate $\hat{r}$, $\hat{\gamma}$, and $\{\hat{k}_{on}^{R}\}$ and using this as input to exact analytical formulas. (**b**) Comparison between the inferred values of $\frac{\hat{r}}{\hat{\gamma}}$ using DGA with experimentally measured values of this parameter from *Jones et al., 2014*. (**c**) Inferred $\hat{k}_{on}^{R}$ values as a function of the mean mRNA level.

chosen because, at its minimum, $\langle \hat{m} \rangle_i = \langle m \rangle_i$ and $\hat{\sigma}_i^m = \sqrt{f_i^m \langle m \rangle_i}$ for all $i$. As above, we set $k_{off}^R = 1$, and focus on estimating the other three parameters $\{r, \gamma, k_{on}^R\}$. When performing our gradient-based optimization, we assume that the transcription rate $r$ and the mRNA degradation rate $\gamma$ are the same for all data points $i$, while allowing $k_{on}^R$ to vary across data points $i$. This reflects the fact that $k_{on}^R$ is a function of the lac repressor concentration which, by design, is varied across data points (see Appendix 6 for details on how this optimization is implemented and calculation of error bars).

The results of this procedure are summarized in *Figure 7*. We find that for the *lac*UD5 promoter $\hat{r} = 90.25$, $\hat{\gamma} = 6.20$ and that $\hat{k}_{on}^R$ varies from a minimum value of 0.18 to a maximum value of 99.0. For the 5DL1 promoters $\hat{r} = 87.48$ and $\hat{\gamma} = 9.80$ and $\hat{k}_{on}^R$ varies between 3.64 and 99.0. Recall that we have normalized all rates to the repressor unbinding rate $k_{off}^R = 1$. These values indicate that mRNA transcription occurs much faster compared to the unbinding of the repressor, suggesting that once the promoter is in an active state, it produces mRNA rapidly. The relatively high mRNA degradation rates indicate a mechanism for fine-tuning gene expression levels, ensuring that mRNA does not persist too long in the cell, which could otherwise lead to prolonged expression even after promoter deactivation.

As expected, the repressor binding rates decrease with the mean mRNA level (see *Figure 7c*). The broad range of repressor binding rates shows that the system can adjust its sensitivity to repressor concentration, allowing for both tight repression and rapid activation depending on the cellular context.

*Figure 7a* shows a comparison between the predictions of the DGA (solid curves) and the experimental data (squares) for mean mRNA levels and the Fano factor . The theoretical curves are obtained by using analytical expression for and from *Gillespie, 2007* with parameters estimated from the DGA. We find that for the lacUD5 and the 5DL1 promoters, the mean percentage errors for predictions of the Fano factor are 25% and 28%, respectively (see Appendix 6).

An appealing feature of *Jones et al., 2014* is that the authors performed independent experiments to directly measure the normalized transcription rate $r/\gamma$ (namely the ratio of the transcription rate and the mRNA degradation rate). This allows us to compare the DGA predictions for these parameters to ground truth measurements of kinetic parameters. In *Figure 7b*, the predictions of the DGA agree remarkably well for both the *lac*UD5 and 5DL1 promoters.

## Designing gene regulatory circuits with desired behaviors

Another interesting application of the DGA is to design stochastic chemical or biological networks that exhibit a particular behavior. In many cases, this design problem can be reformulated as identifying choices of parameter that give rise to a desired behavior. Here, we show that the DGA is ideally suited for such a task. We focus on designing the input–output relation of a four state promoter model of gene regulation (*Lammers et al., 2023*). We have chosen this more complex promoter architecture because, unlike the two-state promoter model analyzed above, it allows for nonequilibrium currents.

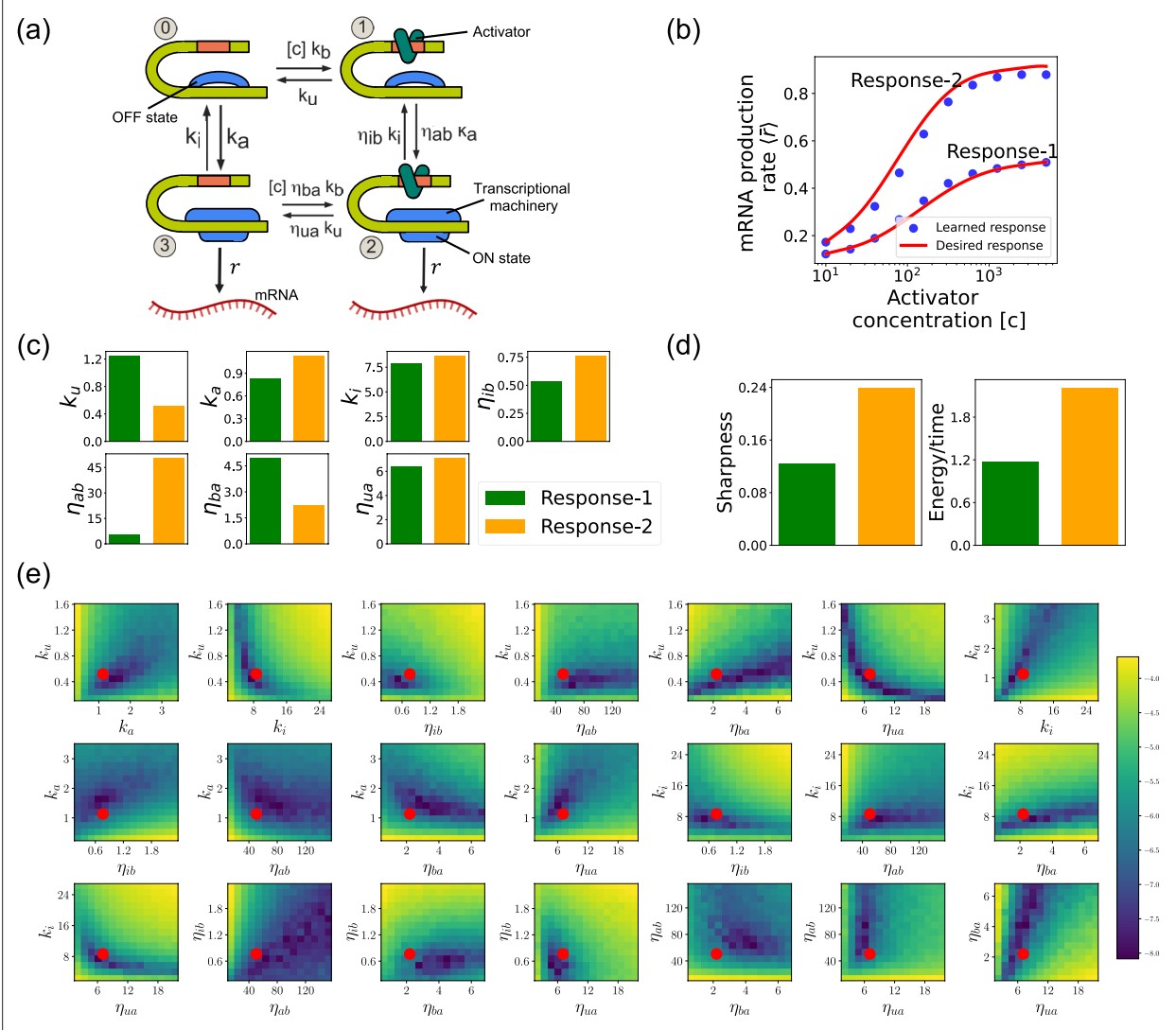

**Figure 8.** Design of the four-state promoter architecture using the differentiable Gillespie algorithm (DGA). (**a**) Schematic of four-state promoter model. (**b**) Target input–output relationships (solid curves) and learned input–output relationships (blue dots) between activator concentration [*c*] and average mRNA production rate. (**c**) Parameters learned by DGA for the two responses in (**b**). (**d**) The sharpness of the response $\frac{d\langle\bar{r}\rangle}{d[c]}[c]$, and the energy dissipated per unit time for two responses in (**b**). (**e**) Logarithm of the loss function for the learned parameter set for Response-2, revealing directions (or curves) of insensitivity in the model's parameter space. The red circles are the learned parameter values.

In making this choice, we are inspired by numerous recent works have investigated how cells can tune kinetic parameters to operate out of equilibrium in order to achieve increased sharpness/sensitivity (***Nicholson and Gingrich, 2023 Lammers et al., 2023***; ***Zoller et al., 2022***; ***Wong and Gunawardena, 2020***; ***Dixit et al., 2024***).

## Model of nonequilibrium promoter

We focus on designing the steady-state input–output relationship of the four-state promoter model of gene regulation model shown in ***Figure 8a***; ***Lammers et al., 2023***. The locus can be in either an 'ON' state where mRNA is transcribed at a rate *r* or an 'OFF' state where the locus is closed and there is no transcription. In addition, a transcription factor (assumed to be an activator) with concentration [*c*] can bind to the locus with a concentration dependent rate [*c*]$k_b$ in the 'OFF' state and a rate [*c*]$\eta_{ba}k_b$ in the 'ON' rate. The activator can also unbind at a rate $k_u$ in the 'OFF' state and a rate $\eta_{ua}k_u$ in the 'ON' state. The average mRNA production rate (averaged over many samples) in this model is given by

$$\langle \bar{r} \rangle = r(\pi_2 + \pi_3) \tag{15}$$

where $\pi_s$ ($s = 2, 3$) is the steady-state probability of finding the system in each of the 'ON' states (see *Figure 8a*).

Such promoter architectures are often studied in the context of protein gradient-based development (*Lammers et al., 2023*; *Estrada et al., 2016*; *Owen and Horowitz, 2023*). One well-known example of such a gradient is the dorsal protein gradient in *Drosophila*, which plays a crucial role in determining the spatial boundaries of gene expression domains during early embryonic development. In this context, the sharpness of the response as a function of activator concentration is a critical aspect. High sharpness ensures that the transition between different gene expression domains occurs over a very narrow region, leading to well-defined and precise boundaries. Inspired by this, our objective is to determine the parameters such that the variation in $\langle \bar{r} \rangle$ as a function of the activator concentration $[c]$ follows a desired response. We consider the two target responses (shown in *Figure 8b*) of differing sharpness, which following *Lammers et al., 2023* we quantify as $\max \left( \frac{\partial \langle \bar{r} \rangle}{\partial [c]} [c] \right)$. For simplicity, we use sixth-degree polynomials to model the input–output functions, with the *x*-axis plotted on a logarithmic scale. We note that our results do not depend on this choice and any other functional form works equally well.

## Loss function

In order to use the DGA to learn a desired input–output relation, we must specify a loss function that quantifies the discrepancy between the desired and actual responses of the promoter network. To construct such a loss function, we begin by discretizing the activator concentration into $N = 10$ logarithmically spaced points, $[c]_i$, where $i = 1, 2, \ldots, N$. For each $[c]_i$, we denote the corresponding average mRNA production rate $\langle \bar{r} \rangle_i$ (see *Equation 15*). After discretization, the loss function is simply the square error between the desired response, $\langle \bar{r} \rangle_i$, and the current response, $\langle \hat{\bar{r}}(\boldsymbol{\theta}) \rangle_i$, of the circuit

$$L = \sum_{i=1}^{N} (\langle \hat{\bar{r}}(\boldsymbol{\theta}) \rangle_i - \langle \bar{r} \rangle_i)^2, \tag{16}$$

where $\langle \hat{\bar{r}}(\boldsymbol{\theta}) \rangle_i$ denotes the predicted average mRNA production rates obtained from the DGA simulations given the current parameters $\boldsymbol{\theta}$. To compute $\langle \hat{\bar{r}} \rangle_i$ for a concentration $[c]_i$, we perform $n = 600$ DGA simulations (indexed by capital letters $A = 1, \ldots, n$) using the DGA and use these simulations to calculate the fraction of time spent in transcriptionally active states (states $s = 2$ and $s = 3$ in *Figure 8a*). If we denote the fraction of time spent in state $s$ in simulation $A$ by $w_s^A$, then we can calculate the probability $\pi_s$ of being in state $s$ by

$$\pi_s = \frac{1}{n} \sum_{A=1}^{n} w_s^A \tag{17}$$

and use *Equation 15* to calculate $\langle \hat{\bar{r}}(\boldsymbol{\theta}) \rangle_i$

As before, we optimize this loss using gradient descent (see *Figure 2*). We assume that the transcription rate $r$ is known (this just corresponds to an overall scaling of mRNA numbers). Since we are concerned only with steady-state properties, we fix the activator binding rate to a constant value, $k_b = 0.02$. This is equivalent to measuring time in units of $k_b^{-1}$. We then use gradient descent to optimize the remaining seven parameters governing transitions between promoter states.

## Assessing circuits found by the DGA

*Figure 8b* shows a comparison between the desired and learned input–output relations. This is good agreement between the learned and desired responses, showing that the DGA is able to design dose–response curves with different sensitivities and maximal values. *Figure 8c* shows the learned parameters for both response curves. Notably, the degree of activation resulting from transcription factor binding, denoted by $\eta_{ab}$, is substantially higher for the sharper response (Response-2). In contrast, the influence on transcription factor binding due to activation, represented by $\eta_{ba}$, is reduced for the sharper response curve. Additionally, the unbinding rate $k_u$ is observed to be lower for the sharper response. However, it is essential to approach these findings with caution, as the parameters are highly interdependent. These interdependencies can be visualized by plotting the loss

function around the optimized parameter values. *Figure 8e* shows two dimensional heat maps of the loss function for Response-2. There are seven free parameters, resulting in a total of 21 possible 2D slices of the loss function within the seven-dimensional loss landscape.

The most striking feature of these plots is the central role played by the parameters $\eta_{ab}$ and $\eta_{ua}$ which must both be high, suggesting that the sharpness in Response-2 may result from creating a high-flux nonequilibrium cycle through the four promoter states (see *Figure 8a*). This observation is consistent with recent works suggesting that creating such nonequilibrium kinetics represents a general design principle for engineering sharp responses (*Lammers et al., 2023*; *Zoller et al., 2022*; *Wong and Gunawardena, 2020*; *Dixit et al., 2024*). To better understand if this is indeed what is happening, we quantified the energy dissipation per unit time (power consumption), $\Phi$, in the nonequilibrium circuit. The energetic cost of operating biochemical networks can be quantified using ideas from nonequilibrium thermodynamics using a generalized Ohm's law of the form (*Lammers et al., 2023*; *Qian, 2007*; *Mehta and Schwab, 2012*; *Lan et al., 2012*; *Lang et al., 2014*; *Mehta et al., 2016*)

$$\Phi = J\Delta\mu \tag{18}$$

where we have defined a nonequilibrium drive

$$\Delta\mu = \ln\left(\frac{\eta_{ab}\eta_{ua}}{\eta_{ib}\eta_{ba}}\right) \tag{19}$$

and the nonequilibrium flux

$$J = \pi_0 k_b [c] - \pi_1 k_u, \tag{20}$$

where $\pi_0$ and $\pi_1$ are the probabilities of finding the system in state 0 and 1, respectively. *Figure 8d* shows a comparison between energy consumption and sharpness of the two learned circuits. Consistent with the results of *Lammers et al., 2023*, we find that the sharper response curve is achieved by consuming more energy.

## Discussion

In this paper, we introduced a fully differentiable variant of the Gillespie algorithm, the DGA. By integrating differentiable components into the traditional Gillespie algorithm, the DGA facilitates the use of gradient-based optimization techniques, such as gradient descent, for parameter estimation and network design. The ability to smoothly approximate the discrete operations of the traditional Gillespie algorithm with continuous functions facilitates the computation of gradients via both forward- and reverse-mode automatic differentiation, foundational techniques in machine learning, and has the potential to significantly expand the utility of stochastic simulations. Our work demonstrates the efficacy of the DGA through various applications, including parameter learning and the design of simple gene regulatory networks.

We benchmarked the DGA's ability to accurately replicate the results of the exact Gillespie algorithm through simulations on a two-state promoter architecture. We found the DGA could accurately approximate the moments of the steady-state distribution and other major qualitative features. Unsurprisingly, it was less accurate at capturing information about the tails of distributions. We then demonstrated that the DGA could be accurately used for parameter estimation on both simulated and real experimental data. This capability to infer kinetic parameters from noisy experimental data underscores the robustness of the DGA, making it a potentially powerful computation tool for real-world applications in quantitative biology. Furthermore, we showcased the DGA's application in designing biological networks. Specifically, for a complex four-state promoter architecture, we learned parameters that enable the gene regulation network to produce desired input–output relationships. This demonstrates how the DGA can be used to rapidly design complex biological systems with specific behaviors. We expect computational design of synthetic circuits with differentiable simulations to become an increasingly important tool in synthetic biology.

There remains much work still to be done. In this paper, we focused almost entirely on properties of the steady states. However, a powerful aspect of the traditional Gillespie algorithm is that it can be used to simulate dynamical trajectories. How to adopt the DGA to utilize dynamical data remains an

extremely important open question. In addition, it will be interesting to see if the DGA can be adapted to understand the kinetic of rare events. It will also be interesting to compare the DGA with other recently developed approximation methods such as those based on tensor networks (*Strand et al., 2022*; *Nicholson and Gingrich, 2023*). Beyond the gene regulatory networks, extending the DGA to handle larger and more diverse datasets will be crucial for applications in epidemiology, evolution, ecology, and neuroscience. On a technical level, this may be facilitated by developing more sophisticated smoothing functions and adaptive algorithms to improve numerical stability and convergence.

The DGA could also be extended to stochastic spatial systems by incorporating reaction–diffusion master equations or lattice-based models. Its differentiability may enable efficient optimization of spatially heterogeneous reaction parameters. However, such extensions may need to address computational scalability and stability in high-dimensional spaces, especially in processes such as diffusion-driven pattern formation or spatial gene regulation.

## Materials and methods

A detailed explanation of how the DGA is implemented using PyTorch is given in the Appendix. In addition, all code for the DGA is available on Github at our Github repository https://github.com/Emergent-Behaviors-in-Biology/Differentiable-Gillespie-Algorithm (copy archived at *Rijal, 2025*).

## Acknowledgements

This work was supported by NIH NIGMS R35GM119461 to PM and Chan-Zuckerburg Institute Investigator grant to PM. The authors also acknowledge support from the Shared Computing Cluster (SCC) administered by Boston University Research Computing Services. We would also like to thank the Mehta and Kondev groups for useful discussions.

## Additional information

### Funding

| Funder | Grant reference number | Author |
| --- | --- | --- |
| National Institute of General Medical Sciences | R35GM119461 | Pankaj Mehta |
| Chan Zuckerberg Initiative | Investigator grant | Pankaj Mehta |

The funders had no role in study design, data collection, and interpretation, or the decision to submit the work for publication.

### Author contributions

Krishna Rijal, Conceptualization, Data curation, Formal analysis, Validation, Investigation, Visualization, Methodology, Writing – original draft; Pankaj Mehta, Conceptualization, Supervision, Funding acquisition, Validation, Investigation, Methodology, Project administration, Writing – review and editing

### Author ORCIDs

Krishna Rijal ⬥ https://orcid.org/0000-0001-7236-7387
Pankaj Mehta ⬥ https://orcid.org/0000-0003-1290-5897

Reviewer #1 (Public review): https://doi.org/10.7554/eLife.103877.3.sa1
Reviewer #2 (Public review): https://doi.org/10.7554/eLife.103877.3.sa2
Reviewer #3 (Public review): https://doi.org/10.7554/eLife.103877.3.sa3
Author response https://doi.org/10.7554/eLife.103877.3.sa4

## Additional files

### Supplementary files
MDAR checklist

### Data availability
All code for the Differentiable Gillespie Algorithm is freely available on GitHub at https://github.com/Emergent-Behaviors-in-Biology/Differentiable-Gillespie-Algorithm (copy archived at *Rijal, 2025*). The study uses previously published experimental datasets from *Jones et al., 2014*. This dataset is publicly available through Science at https://doi.org/10.1126/science.1255301. No additional proprietary or restricted datasets were used in this study.

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

## Appendix 1

### Balancing accuracy and stability: hyperparameter tradeoffs

In this section, we explore the tradeoffs involved in tuning the hyperparameters $a$ and $b$ in the DGA. These hyperparameters are crucial for balancing the accuracy and numerical stability of the DGA in approximating the exact Gillespie algorithm.

The hyperparameter $a^{-1}$ controls the steepness of the sigmoid function used to approximate the Heaviside step function in reaction selection. Similarly, $b^{-1}$ determines the sharpness of the Gaussian function used to approximate the Kronecker delta function in abundance updates. A larger value of $a^{-1}$ or $b^{-1}$ results in a steeper sigmoid or Gaussian function, thus more closely approximating the discrete functions in the exact Gillespie algorithm.

### Accuracy of the forward DGA simulations

To assess the impact of these hyperparameters on the accuracy of the DGA, we measure the ratio of the Jensen–Shannon divergence between the DGA-generated PDF $p_{\text{DGA}}$ and the exact PDF $p_{\text{exact}}$, normalized by the entropy $\text{H}(p_{\text{exact}})$ of the exact steady-state PDF (see *Equation 9*). This ratio, $\frac{\text{JSD}(p_{\text{DGA}} \| p_{\text{exact}})}{\text{H}(p_{\text{exact}})}$, provides a measure of how closely the DGA approximates the exact Gillespie algorithm.

*Appendix 1—figure 1* shows the ratio $\frac{\text{JSD}}{\text{H}}$ as a function of the sharpness parameters $a^{-1}$ and $b^{-1}$. In panel (a), $b^{-1}$ is fixed at 20, and $a^{-1}$ is varied. In panel (b), $a^{-1}$ is fixed at 200, and $b^{-1}$ is varied. Some key insights can be drawn from these plots.

First, as $a^{-1}$ or $b^{-1}$ increases, the ratio $\frac{\text{JSD}}{\text{H}}$ decreases, indicating that the DGA's approximation becomes more accurate. This is because steeper sigmoid and Gaussian functions better mimics the discrete steps of the exact Gillespie algorithm. Interestingly, while the ratio decreases for both parameters, $a^{-1}$ plateaus at high values, whereas $b^{-1}$ rises at high values. This plateau occurs because the sigmoid function used for reaction selection becomes so steep that it effectively becomes a step function, beyond which further steepening has negligible impact. As $b^{-1}$ becomes very large, the width of the Gaussian function used for abundance updates becomes extremely narrow. In such a scenario, it becomes increasingly improbable for the chosen reaction index to fall within this narrow width, especially because the reaction indices are not exact integers but are instead near-integer continuous values. Therefore, as $b^{-1}$ becomes very large, the discrepancy between the DGA-generated probabilities and the exact probabilities widens, causing the ratio $\frac{\text{JSD}}{\text{H}}$ to increase.

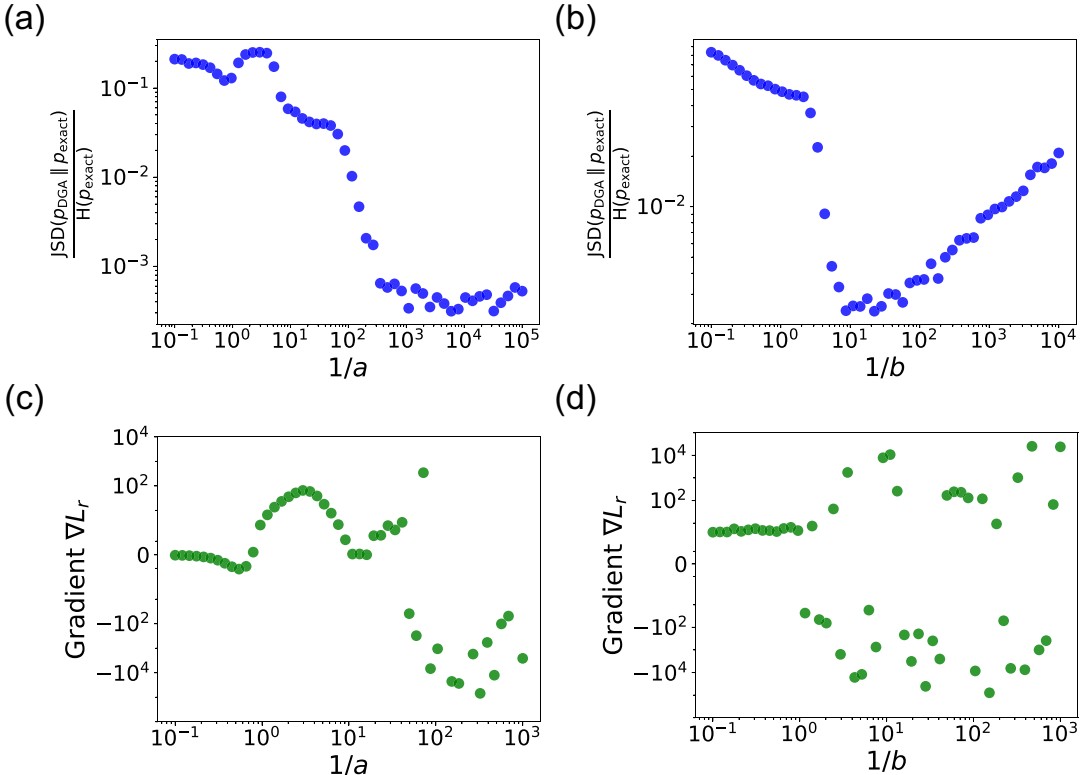

**Appendix 1—figure 1.** In panels (**a**) and (**b**), we plot the ratio of the Jensen–Shannon divergence $\text{JSD}\left(p_{\text{DGA}}\|p_{\text{exact}}^{\text{ss}}\right)$ between the differentiable Gillespie PDF $p_{\text{DGA}}$ and the exact steady-state PDF $p_{\text{exact}}^{\text{ss}}$, and the Shannon entropy $\text{H}(p_{\text{exact}}^{\text{ss}})$ of the exact steady-state PDF, as a function of the two sharpness parameters $1/a$ and $1/b$. In panel (**a**), $1/b = 20$; in panel (**b**), $1/a = 200$. The simulation time is set to 10. In panels (**c**) and (**d**), for these same values, we show the gradient $\nabla L_r$ of the loss function $L$ with respect to the parameter $r$ near the true parameter values. In all the plots, the values of the rates are: $k_{\text{on}}^{\text{R}} = 0.5$, $k_{\text{off}}^{\text{R}} = 1$, $r = 10$, and $\gamma = 1$. 5000 trajectories are used to obtain the PDFs, while 200 trajectories are used to obtain the gradients.

## Stability of the backpropagation of DGA simulations

Numerical stability for gradient computation is crucial. Therefore, it is important to examine how the gradient behaves as a function of the hyperparameters. Panels (**c**) and (**d**) of *Appendix 1—figure 1* provide insights into this behavior. The plots show the gradient $\nabla L_r$ of the loss function $L$ with respect to the parameter $r$ near the true parameter values. The gradients are computed for the loss function in *Equation 11* with $\langle m \rangle$ and $\sigma_m$ equal to 8 and 2.5, respectively, for the parameter values $k_{\text{on}}^{\text{R}} = 0.5$, $k_{\text{off}}^{\text{R}} = 1$, $r = 10$, and $\gamma = 1$. With these parameter values, the true mean and standard deviation are equal to 6.67 and 3.94, respectively.

As $a^{-1}$ or $b^{-1}$ increases, the gradients become more accurate, but their numerical stability can be compromised. This is evidenced by the increased variability and erratic behavior in the gradients at very high sharpness values (see *Appendix 1—figure 1*). Hence, very large values of $a^{-1}$ or $b^{-1}$ lead to oscillations and convergence issues, highlighting the need for a balance between accuracy and stability.

The tradeoff between accuracy and numerical stability is evident. This necessitates careful tuning of $a$ and $b$ to ensure stable and efficient optimization. In practice, this involves selecting values that provide sufficient approximation quality without compromising the stability of the gradients.

## Appendix 2

### Implementation of the DGA in PyTorch

This section explains the implementation of the DGA used to simulate the two-state promoter model. The implementation can be adapted to any model where the stoichiometric matrix and propensities are known. The model involves promoter state switching and mRNA production/degradation (*Figure 3a*), and the algorithm is designed to handle these sub-processes effectively.

### Stoichiometric matrix:

The stoichiometric matrix is a key component in modeling state changes due to reactions. Each row represents a reaction, and each column corresponds to a state variable (promoter state and mRNA level). The matrix for the two-state promoter model includes:

- Reaction 1: Promoter state transitions from the OFF state (–1) to the ON state (+1).
- Reaction 2: Production of mRNA.
- Reaction 3: Promoter state transitions from the ON state (+1) to the OFF state (–1).
- Reaction 4: Degradation of mRNA.

The stoichiometric matrix $S$ for this model is:

$$S = \begin{pmatrix} 2 & 0 \\ 0 & 1 \\ -2 & 0 \\ 0 & -1 \end{pmatrix}$$

In this matrix, the rows correspond to the reactions listed above, and the columns represent the promoter state and mRNA level, respectively.

### Propensity calculations:

The levels vector is a two-dimensional vector where the first element represents the promoter state (–1 or +1) and the second element represents the mRNA number $m$. Propensities are the rates at which reactions occur, given the current state of the system. In our PyTorch implementation, the propensities are calculated using the following expressions:

```
propensities = torch.stack([
    kon * torch.sigmoid(-c * levels[0]),
    # Promoter state switching from -1 to +1
    r * torch.sigmoid(-c * levels[0]),
    # mRNA production
    torch.sigmoid(c * levels[0]),
    # Promoter state switching from +1 to -1
    g * levels [1]
    # mRNA degradation
])
```

Each propensity corresponds to a different reaction:

- Promoter state switching from –1 to +1: The value of `kon * torch.sigmoid(-c * levels[0])` is around $k_{on}^{R}$ when the `levels[0]` (promoter state) is around –1 and close to zero when `levels[0]` is around +1. The sigmoid function ensures a smooth transition, allowing differentiability and preventing abrupt changes in rates. The constant $c$ controls the sharpness of the sigmoid function.
- mRNA production: The rate is proportional to $r$ and modulated by the same sigmoid function, `torch.sigmoid(-c * levels[0])`, such that the rate is equal to $r$ only when the promoter is in –1 state.
- Promoter state switching from +1 to –1: The rate is set to 1 and modulated by the sigmoid function, `torch.sigmoid(c * levels[0])`, such that the rate is equal to 1 only when the promoter is in +1 state.

- mRNA degradation: The rate is proportional to the current mRNA level, `g * levels [1]`, reflecting the natural decay of mRNA with rate $m\gamma$.

Using the sigmoid function in the propensity calculations is crucial for ensuring smooth and differentiable transitions between states. This smoothness is essential for gradient-based optimization methods, which rely on continuous and differentiable functions to compute gradients effectively. Without the sigmoid function, the propensity rates could change abruptly, leading to numerical instability and difficulties in optimizing the model parameters.

Reaction selection function: We define a function `reaction_selection` that selects the next reaction to occur based on the transition points and a random number between [0,1]. The function basically implements using *Equation 6*. The transition points are first calculated from the cumulative sum of reaction rates normalized to the total rate.

Gillespie simulation function: The main `gillespie_simulation` function uses the previously described functions, each with specific roles, to perform the actual simulation step-by-step, as shown in *Figure 1*. This function iterates through the number of simulations, updating the system's state and the propensities of each reaction at each step.

State jump function: We define another function `state_jump` that calculates the state change vector when a reaction occurs. It uses a Gaussian function to smoothly transition between states based on the selected reaction index and the stoichiometry matrix (see *Equation 8*).

## Appendix 3

### Estimating confidence intervals for parameters

In this section, we describe the methodology used to estimate the confidence intervals for the parameters using polynomial fitting and numerical techniques. This approach uses the results of multiple simulations to determine the range within which the parameter $\theta_i$ is likely to lie, based on the curvature of the loss function around its minimum value. Specifically, we perform the following steps:

1. Parameter initialization: Set up and initialize the necessary parameters. This includes defining the number of evaluation points, the number of simulations, the simulation time, and the hyper-parameters $a^{-1}$, $b^{-1}$, and $c$.

2. Range generation: For each set of learned parameters $\hat{\boldsymbol{\theta}}$, generate a range of values for the parameter of interest $\theta_i$ while keeping the other parameters fixed at their learned values. Let the learned value of the parameter $\theta_i$ be $\hat{\theta}_i$. Then the range of evaluation is approximately $[0.2\hat{\theta}_i, 2\hat{\theta}_i]$, depending on the flatness of the loss landscape around its minimum.

3. Simulation and loss calculation: For each value in this range, perform forward DGA simulations to calculate the mean and standard deviation of the results, using which the loss function is computed and stored.

4. Polynomial fitting: Fit a polynomial of degree 6 to the computed loss values across the range of $\theta_i$. Compute the first and second derivatives of the fitted polynomial to identify the minimum and evaluate the curvature.

5. Identifying minimum: Identify the valid minimum $\theta_i^{\min}$ of the loss function by solving for the roots of the first derivative and filtering out the points where the second derivative is positive.

6. Curvature and standard deviation calculation: Calculate the curvature of the loss landscape at its minimum using its second derivative at the minimum. An estimation of the standard deviation $\delta_{\theta_i}$ of the parameter $\theta_i$ is given by:

$$\delta_{\theta_i} = \left( \sqrt{\frac{\partial^2 L}{\partial \theta_i^2}} \right)^{-1} \Bigg|_{\theta_i = \theta_i^{\min}} \tag{C1}$$

7. Confidence interval calculation: The loss function is typically asymmetric around its minimum, with the left side usually steeper than the right (see **Appendix 3—figure 1**). To determine the right error bar, we use $\theta_i^{\min} + 1.96\delta_{\theta_i}$. For the left error bar, we find the point where $L(\theta_i^{\min} - \delta) = L(\theta_i^{\min} + 1.96\delta_{\theta_i})$ (see **Appendix 3—figure 1**). Therefore, the balanced 95% CI for the parameter $\theta_i$ is given by:

$$CI_{\theta_i} = [\theta_i^{\min} - \delta, \theta_i^{\min} + 1.96\delta_{\theta_i}] \tag{C2}$$

This methodology allows for a robust estimation of the confidence intervals, providing insights into the reliability and precision of the parameter estimates.

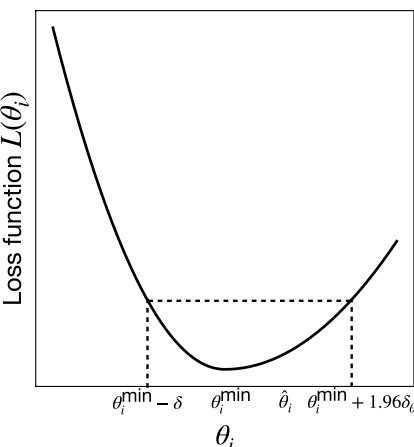

**Appendix 3—figure 1.** Error bars estimation for asymmetric loss function.

## Appendix 4

### Demonstrating parameter degeneracy in the two-state promoter model

We will now demonstrate the existence of degeneracy in the two-state promoter architecture. Setting $k_{\text{off}}^{\text{R}} = 1$ in the analytical expressions for the mean $\langle m \rangle$ and the Fano factor $f$ from *Jones et al., 2014*, we have:

$$\langle m \rangle = \frac{1}{k_{\text{on}}^{\text{R}} + 1} \cdot \frac{r}{\gamma},$$

$$f = 1 + \frac{k_{\text{on}}^{\text{R}}}{k_{\text{on}}^{\text{R}} + 1} \cdot \frac{r}{k_{\text{on}}^{\text{R}} + 1 + \gamma} \tag{D1}$$

We want to solve *Equation D1* for $r$ and $\gamma$. By isolating $r$ in the expression for $\langle m \rangle$, we obtain:

$$r = \langle m \rangle \cdot \gamma \cdot (k_{\text{on}}^{\text{R}} + 1) \tag{D2}$$

Next, we substitute the expression for $r$ from *Equation D2* into the expression for the $f$ in *Equation D1*:

$$f = 1 + \frac{k_{\text{on}}^{\text{R}}}{k_{\text{on}}^{\text{R}} + 1} \cdot \frac{\langle m \rangle \cdot \gamma \cdot (k_{\text{on}}^{\text{R}} + 1)}{k_{\text{on}}^{\text{R}} + 1 + \gamma}$$

$$= 1 + \frac{k_{\text{on}}^{\text{R}} \cdot \langle m \rangle \cdot \gamma}{k_{\text{on}}^{\text{R}} + 1 + \gamma}$$

$$\Rightarrow (f - 1) \cdot (k_{\text{on}}^{\text{R}} + 1 + \gamma) = k_{\text{on}}^{\text{R}} \cdot \langle m \rangle \cdot \gamma$$

Expanding and isolating terms involving $\gamma$:

$$(f - 1) \cdot k_{\text{on}}^{\text{R}} + (f - 1) + (f - 1) \cdot \gamma = k_{\text{on}}^{\text{R}} \cdot \langle m \rangle \cdot \gamma$$

Rearranging to solve for $\gamma$:

$$(f - 1) + (f - 1) \cdot k_{\text{on}}^{\text{R}} = \gamma \cdot (k_{\text{on}}^{\text{R}} \cdot \langle m \rangle - f + 1)$$

Thus, we obtain:

$$\gamma = \frac{(f - 1) \cdot (k_{\text{on}}^{\text{R}} + 1)}{k_{\text{on}}^{\text{R}} \cdot \langle m \rangle - f + 1} \tag{D3}$$

We substitute back *Equation 28* in *Equation 23* to obtain $r$:

$$r = \frac{\langle m \rangle (f - 1) \cdot (k_{\text{on}}^{\text{R}} + 1)^2}{k_{\text{on}}^{\text{R}} \cdot \langle m \rangle - f + 1} \tag{D4}$$

*Equations D3 and D4* indicate that for any given $k_{\text{on}}^{\text{R}}$, there exists a corresponding pair of values for $r$ and $\gamma$ that satisfy the equations. Therefore, the solutions are not unique, demonstrating a high degree of degeneracy in the parameter space. This degeneracy arises because multiple combinations of $\{r, \gamma, k_{\text{on}}^{\text{R}}\}$ can produce the same observable $\langle m \rangle$ and $f$. The lack of unique solutions is a common issue in parameter estimation for complex systems, where different parameter sets can lead to similar system behaviors.

### Resolving degeneracy with known $\gamma$

To resolve the degeneracy, we can fix the value of $\gamma$. Now, if the rate $\gamma$ is known, we can solve *Equation D1* for the other parameters. Starting by rearranging the mean expression from *Equation D1*:

$$k_{\text{on}}^{\text{R}} = \frac{r}{\gamma \cdot \langle m \rangle} - 1 \tag{D5}$$

Substituting *Equation D5* in the expression of $f$ in *Equation D1*, we have

$$f = 1 + \frac{r - \gamma\langle m\rangle}{r} \cdot \frac{r\gamma\langle m\rangle}{r + \gamma^2\langle m\rangle}$$

$$\Rightarrow (f - 1)(r + \gamma^2\langle m\rangle) = (r - \gamma\langle m\rangle)(\gamma\langle m\rangle)$$

$$\Rightarrow r(f - 1 - \gamma\langle m\rangle) = -\gamma^2\langle m\rangle^2 - (f - 1)\gamma^2\langle m\rangle$$ 
(D6)

$$\Rightarrow r = \frac{\langle m\rangle\gamma^2(\langle m\rangle + f - 1)}{\gamma\langle m\rangle - f + 1}$$

Substituting *Equation D6* into *Equation D5*, we obtain:

$$k_{on}^{R} = \frac{\gamma(\langle m\rangle + f - 1)}{\gamma\langle m\rangle - f + 1} - 1$$ 
(D7)

*Equations D6 and D7* provide unique solutions for $r$ and $k_{on}^{R}$ given a known value of $\gamma$. By fixing $\gamma$, the degeneracy is resolved, and the remaining parameters can be accurately determined.

## Appendix 5

### Estimating confidence intervals for $\langle m \rangle$ and $\sigma$

To assess the reliability of the statistics predicted through DGA-based optimization, we calculate error bars using the following procedure. For the non-degenerate situation, we use the learned parameter values $\hat{k}_{\mathrm{on}}^{\mathrm{R}}$ and $\hat{r}$, and perform forward DGA simulations with many different random seeds. This generates multiple samples of the mean mRNA level $\langle m \rangle$ and the standard deviation $\sigma_m$. These samples provide us with the variability due to the stochastic nature of the simulations. The 95% CIs are then determined using their standard deviations, denoted as $\delta_{\langle m \rangle}$ and $\delta_{\sigma_m}$. For the mean mRNA level, the CI is calculated as:

$$\mathrm{CI}_{\langle m \rangle} = [\hat{\langle m \rangle} - 1.96 \times \delta_{\langle m \rangle}, \hat{\langle m \rangle} + 1.96 \times \delta_{\langle m \rangle}] \tag{E1}$$

For the standard deviation of mRNA levels, the CI is calculated as:

$$\mathrm{CI}_{\sigma_m} = [\hat{\sigma}_m - 1.96 \times \delta_{\sigma_m}, \hat{\sigma}_m + 1.96 \times \delta_{\sigma_m}] \tag{E2}$$

## Appendix 6

## DGA-based optimization for experimental data and estimation of errors

### Optimization procedure

Given a set of measurements of the mean mRNA expression levels ($\langle m \rangle_i$) and the Fano factor ($f_i^m$) for promoters (*lac*UD5 and 5DL1), we construct a loss function as follows:

$$L = \sum_{i=1}^{N} (\langle \hat{m} \rangle_i - \langle m \rangle_i)^2 + \sum_{i=1}^{N} (\hat{\sigma}_i^m - \sqrt{f_i^m \langle m \rangle_i})^2 \tag{F1}$$

where $i$ runs over data points (each with a different lac repressor concentration), and $\langle \hat{m} \rangle_i$ and $\hat{\sigma}_i^m$ are the mean and standard deviation obtained from a sample of DGA simulations. This loss function is chosen because, at its minimum, $\langle \hat{m} \rangle_i = \langle m \rangle_i$ and $\hat{\sigma}_i^m = \sqrt{f_i^m \langle m \rangle_i}$ for all $i$.

For the optimization, we set $k_{\text{off}}^{\text{R}} = 1$ and focus on estimating the parameters $\{r, \gamma, k_{\text{on}}^{\text{R}}\}$. During the gradient-based optimization, the transcription rate $r$ and the mRNA degradation rate $\gamma$ are assumed to be the same for all data points $i$, while allowing $k_{\text{on}}^{\text{R}}$ to vary across data points $i$. This reflects the fact that $k_{\text{on}}^{\text{R}}$ is a function of the lac repressor concentration, which is varied across data points.

Instead of $k_{\text{on}}^{\text{R}}$, we actually learn a transformed parameter $p_{\text{off}} = 1/(1 + k_{\text{on}}^{\text{R}})$, which is the probability for the promoter to be in the OFF state. This approach is based on our observation that the gradient of the loss function with respect to $p_{\text{off}}$ is more numerically stable compared to the gradient with respect to $k_{\text{on}}^{\text{R}}$.

The parameters are initialized randomly as follows:
- $r$ is initialized to a random number in [0, 100].
- $\gamma$ is initialized to a random number between [0, 10].
- The $p_{\text{off}}$ values, which depend on the index $i$ of the data points, are initially set as linearly spaced points within the range [0.03, 0.97].

The hyperparameters used for the simulations are as follows:
- Number of simulations: 200
- Simulation time: 0.2
- Steepness of the sigmoid function: $a^{-1} = 200.0$
- Sharpness of the Gaussian function: $b^{-1} = 20.0$
- Steepness of the sigmoid in propensities: $c = 20.0$

The parameters are iteratively updated to minimize the loss function. During each iteration of the optimization, the following steps are performed:

1. Forward simulation: The DGA is used to simulate the system, generating predictions for the mean mRNA levels and their standard deviations.
2. Loss calculation: The loss function is computed based on the differences between the simulated and experimentally measured values of the mean mRNA levels and standard deviations (see *Equation F1*).
3. Gradient calculation: The gradients of the loss function with respect to the parameters $r$, $\gamma$, and $k_{\text{on}}^{\text{R}}$ are calculated using backpropagation.
4. Parameter update: The ADAM optimizer updates the parameters in the direction that reduces the loss function. ADAM adjusts the learning rates based on the history of gradients and their moments. The learning rate used is 0.1.

The values of the parameters and the loss value are saved after each iteration. The parameter values corresponding to the minimum loss after convergence are picked at the end.

### Goodness of fit

To quantitatively assess the goodness of the fit, we define the mean percentage error (MPE) as follows:

$$\text{MPE} = \frac{100\%}{N} \sum_{i=1}^{N} \frac{\left| f_i^m - \hat{f}_i^m \right|}{f_i^m}, \tag{F2}$$

where $\hat{f}_i^m$ is the predicted Fano factor for the $i$th data point. This metric provides a measure of the average discrepancy between the predicted and experimental Fano factors, expressed as a percentage of the experimental values.

## Error bars

The error bars in the ratio $\hat{r}/\hat{\gamma}$ are obtained by applying error propagation to the standard deviations $\delta_r$ and $\delta_\gamma$ of the individual values $\hat{r}$ and $\hat{\gamma}$. The propagated error is given by:

$$\delta_{\frac{r}{\gamma}} = \frac{\hat{r}}{\hat{\gamma}} \sqrt{\left(\frac{\delta_r}{\hat{r}}\right)^2 + \left(\frac{\delta_\gamma}{\hat{\gamma}}\right)^2}, \tag{F3}$$

where $\delta_r$ and $\delta_\gamma$ are the standard deviations of $\hat{r}$ and $\hat{\gamma}$, respectively. These standard deviations are obtained using the curvature of the loss function, as discussed earlier.

