## [Editor Report · eLife Assessment]

This **important** study introduces a fully differentiable variant of the Gillespie algorithm as an approximate stochastic simulation scheme for complex chemical reaction networks, allowing kinetic parameters to be inferred from empirical measurements of network outputs using gradient descent. The concept and algorithm design are **convincing** and innovative. While the proofs of concept are promising, some questions are left open about implications for more complex systems that cannot be addressed by existing methods. This work has the potential to be of significant interest to a broad audience of quantitative and synthetic biologists.

---

## [Referee Report · Reviewer #1 (Public review)]

Summary:

This work introduces the differentiable Gillespie algorithm, DGA, which is a differentiable variant of the celebrated (and exact) Gillespie algorithm commonly used to perform stochastic simulations across numerous fields, notably in the life sciences. The proposed DGA approximates the exact Gillespie algorithm using smooth functions yielding a suitable approximate differentiable stochastic system as a proxy for the underlying discrete stochastic system, where DGA stochastic reactions have continuous reaction index and the species abundances. To illustrate their methodology, the authors specifically consider in detail the case of a well-studied two-state promoter gene regulation system that they analyze using a machine learning approach, and by combining simulation data with analytical results. For the two-state promoter gene system, the DGA is benchmarked by accurately reproducing the results of the exact Gillespie algorithm. For this same simple system, the authors also show how the DGA can be used for estimating kinetic parameters of both simulated and real noisy experimental data. This lets them argue convincingly that the DGA can become a powerful computation tool for applications in quantitative and synthetic biology. In order to argue that the DGA can be employed to design circuits with ad-hoc input-output relations, these considerations are then extended to a more complex four-state promoter model of gene regulation. The main strength of the paper is its clarity and its pedagogical presentation of the simulation methods.

Strengths:

The main strength of the paper is its clarity and its pedagogical presentation of the simulation methods.

Weaknesses:

It would have been useful to have a brief discussion, based on a concrete example, of what can be achieved with the DGA and is totally beyond the reach of the Gillespie algorithm and the numerous existing stochastic simulation methods. A more comprehensive and quantitative analysis of the limitations of the DGA, e.g. for rare events, and how it might be used for stochastic spatial systems would have also been helpful. However, this is arguably beyond the scope of this study whose primary goal is to introduce the DGA and demonstrate that it can achieve tasks like parameter estimation and network design.

Comments on revisions:

The authors have made a sound effort to address many of the comments raised in the previous reports. This has helped improve the clarity of the discussion.

---

## [Referee Report · Reviewer #2 (Public review)]

Summary:

In this work, the authors present a differentiable version of the widely-used Gillespie Algorithm. The Gillespie Algorithm has been used for decades to simulate the behavior of stochastic biochemical reaction networks. But while the Gillespie Algorithm is a powerful tool for the forward simulation of biochemical systems given some set of known reaction parameters, it cannot be used for reverse process, i.e. inferring reaction parameters given a set of measured system characteristics. The Differentiable Gillespie Algorithm ("DGA") overcomes this limitation by approximating two discontinuous steps in the Gillespie Algorithm with continuous functions. This makes it possible to calculate of gradients for each step in the simulation process which, in turn, allows the reaction parameters to be optimized via powerful backpropagation techniques. In addition to describing the theoretical underpinnings of DGA, the authors demonstrate different potential use-cases for the algorithm in the context of simple models of stochastic gene expression.

Overall, the DGA represents an important conceptual step forward for the field and should lay the groundwork for exciting innovations in the analysis and design of stochastic reaction networks. At the same time, significantly more work is needed to establish when the approximations made by DGA are valid and to demonstrate the viability of the algorithm in the context of complicated reaction networks.

Strengths:

This work makes an important conceptual leap by introducing a version of the Gillespie Algorithm that is end-to-end differentiable. This idea alone has the potential to drive a number of exciting innovations in the analysis, inference, and design of biochemical reaction networks. Beyond the theoretical adjustments, the authors also implement their algorithm in a Python-based codebase that combines DGA powerful optimization libraries like PyTorch. This codebase has the potential to be of interest to a wide range of researchers, even if the true scope of the method's applicability remains to be fully determined.

The authors also demonstrate how DGA can be used in practice both to infer reaction parameters from real experimental data (Figure 7) and to design networks with user-specified input-output characteristics (Figure 8). These illustrations should provide a nice roadmap for researchers interested in applying DGA to their own projects/systems.

Finally, although it does not stem directly from DGA, the exploration of pairwise parameter dependencies in different network architectures provides an interesting window into the design constraints (or lack thereof) that shape the architecture of biochemical reaction networks.

Weaknesses:

While it is clear that the DGA represents an important conceptual advancement, the authors do not do enough in the present manuscript to (i) validate the robustness of DGA inference and (ii) demonstrate that DGA inference works in the kinds of complex biochemical networks where it would actually be of legitimate use.

It is to the authors' credit that they are open and explicit about the potential limitations of DGA due to breakdowns in its continuous approximations. However they do not provide the reader with nearly enough empirical (i.e. simulation-based) or theoretical context to assess when, why, and to what extent DGA will fail in different situations. In Figure 2, they compare DGA to GA (i.e. ground-truth) in the context of a simple two state model of a stochastic transcription. Even in this minimal system, we see that DGA deviates notably from ground-truth both in the simulated mRNA distributions (Figure 2A) and in the ON/OFF state occupancy (Figure 2C). This begs the question of how DGA will scale to more complicated systems, or systems with non-steady state dynamics. Will the deviations become more severe? This is important because, in practice, there is really not much need for using DGA with a simple 2 state system-we have analytic solutions for this case. It is the more complex systems where DGA has the potential to move the needle.

A second concern is that the authors' present approach for parameter inference and error calculation does not seem to be reliable. For example, in Figure 5A, they show DGA inference results for the ON rate of a two-state system. We see substantial inference errors in this case, even though the inference problem should be non-degenerate in this case. One reason for this seems to be that the inference algorithm does not reliably find the global minimum of the loss function (Figure 2B). To turn DGA into a viable approach, it is paramount that the authors find some way to improve this behavior, perhaps by using multiple random initializations to better search the loss space.

Finally, the authors do a good job of illustrating how DGA might be used to infer biological parameters (Figure 7) and design reaction networks with desired input-output characteristics (Figure 8). However, analytic solutions exist for both of the systems they select for examples. This means that, in practice, there would be no need for DGA in these contexts, since one could directly optimize, e.g., the expressions for the mean and Fano Factor of the system in Figure 7A. I still believe that it is useful to have these examples, but it seems critical to add a use-case where DGA is the only option.

Comments on revisions:

I am concerned that the results in Figure 8D may not be correct, or that the authors may be mis-interpreting them. From my reading of the paper they cite (Lammers & Flamholz 2023), the equilibrium sharpness limit for the network they consider in Figure 8 should be 0.25. But both solutions shown in Figure 8D fall below this limit, which means that they have sharpness levels that could have been achieved with no energy expenditure. If this is the case, then it would imply that while both systems do dissipate energy, they are not doing so productively; meaning that the same results could be achieved while holding Phi=0.

I acknowledge that this could be due to a difference in how they measure sharpness, but wanted to raise it here in case it is, in fact, a genuine issue with the analysis.

There should be an easy fix for this: just set the sharper "desired response" curve in 8b to be such that it demands non-equilibrium sharpness levels (0.25)

---

## [Referee Report · Reviewer #3 (Public review)]

Summary:

This manuscript introduces a differentiable variant of the Gillespie algorithm (DGA) that allows gradient calculation using backpropagation. The most significant contribution of this work is the development of the DGA itself, a novel approach to making stochastic simulations differentiable. This is achieved by replacing discontinuous operations in the traditional Gillespie algorithm with smooth, differentiable approximations using sigmoid and Gaussian functions. This conceptual advance opens up new avenues for applying powerful gradient-based optimization techniques, prevalent in machine learning, to studying stochastic biological systems.

The method was tested on a simple two-state promoter model of gene expression. The authors found that the DGA accurately captured the moments of the steady-state distribution and other major qualitative features. However, it was less accurate at capturing information about the distribution's tails, potentially because rare events result from frequent low-probability reaction events where the approximations made by the DGA have a greater impact. The authors also used the DGA to design a four-state promoter model of gene regulation that exhibited a desired input-output relationship. The DGA could learn parameters that produced a sharper response curve, which was achieved by consuming more energy.

The authors conclude that the DGA is a powerful tool for analyzing and designing stochastic systems. The discussion lays several open questions in the field and constructively addresses shortcomings of the proposed method as well as potential ways forward.

Strengths:

The DGA allows gradient-based optimization techniques to estimate parameters and design networks with desired properties.

The DGA efficacy in estimating kinetic parameters from both synthetic and experimental data. This capability highlights the DGA's potential to extract meaningful biophysical parameters from noisy biological data.

The DGA's ability to design a four-state promoter architecture exhibits a desired input-output relationship. This success indicates the potential of the DGA as a valuable tool for synthetic biology, enabling researchers to engineer biological circuits with predefined behaviours.

Weaknesses:

The study primarily focuses on analysing the steady-state properties of stochastic systems.

Comments on revisions:

Thank you for addressing all the points raised. I am looking forward to seeing the next steps in DGAs development and performance!

---

## [Author Response]

The following is the authors’ response to the current reviews.

**Response to Reviewer 2’s comments:**
I am concerned that the results in Figure 8D may not be correct, or that the authors may be mis-interpreting them. From my reading of the paper they cite (Lammers & Flamholz 2023), the equilibrium sharpness limit for the network they consider in Figure 8 should be 0.25. But both solutions shown in Figure 8D fall below this limit, which means that they have sharpness levels that could have been achieved with no energy expenditure. If this is the case, then it would imply that while both systems do dissipate energy, they are not doing so productively; meaning that the same results could be achieved while holding Phi=0.I acknowledge that this could be due to a difference in how they measure sharpness, but wanted to raise it here in case it is, in fact, a genuine issue with the analysis.There should be an easy fix for this: just set the sharper "desired response" curve in 8b to be such that it demands non-equilibrium sharpness levels (0.25<S<0.5).

Thank you for raising this point regarding the interpretation of our results in Figure 8D. We agree that if the equilibrium sharpness limit for this particular network is around 0.25 (as shown by Lammers & Flamholz 2023), then achieving a sharpness below this threshold could, in principle, be accomplished without any energy expenditure. However, in our current design approach, the loss function is solely designed to enforce agreement with a target mean mRNA level at different input concentrations; it does not explicitly constrain energy dissipation, noise, or other metrics. Consequently, the DGA has no built-in incentive to minimize or optimize energy consumption, which means the resulting solutions may dissipate energy without exceeding the equilibrium sharpness limit.

In other words, the same input–output relationship could theoretically be achieved with \Phi = 0 if an explicit constraint or regularization term penalizing energy usage had been included. As noted, adding such a term (e.g., penalizing \Phi^2) is conceptually straightforward but falls outside the scope of this study. Our primary goal is to demonstrate the flexibility of the DGA in designing a desired response, rather than to delve into energy–sharpness trade-offs or other biological considerations

While we appreciate the suggestion to set a higher target sharpness that exceeds the equilibrium limit, we believe the current example effectively demonstrates the DGA’s ability to design circuits with desired input-output relationships, which is the primary focus of this study. Researchers interested in optimizing energy efficiency, burst size, burst frequency, noise, response time, mutual information, or other system properties can easily extend our approach by incorporating additional terms into the loss function to target these specific objectives.

We hope this explanation addresses your concern and clarifies that the manuscript provides sufficient context for readers to interpret the results in Figure 8D correctly.

The following is the authors’ response to the original reviews.

**Reviewer #1 (Public review):**

We thank Reviewer #1 for their thoughtful feedback and appreciation of the manuscript's clarity. Our primary goal is to introduce the DGA as a foundational tool for integrating stochastic simulations with gradient-based optimization. While we recognize the value of providing detailed comparisons with existing methods and a deeper analysis of the DGA’s limitations (such as rare event handling), these topics are beyond the scope of this initial work. Our focus is on presenting the core concept and demonstrating its potential, leaving more extensive evaluations for future research.

**Reviewer #2 (Public review):**

We thank Reviewer #2 for their detailed and constructive feedback. We appreciate the recognition of the DGA as a significant conceptual advancement for stochastic biochemical network analysis and design.

Weaknesses:(1) Validation of DGA robustness in complex systems:

Our primary goal is to introduce the DGA framework and demonstrate its feasibility. While validation on high-dimensional and non-steady-state systems is important, it is beyond the scope of this initial work. Future studies may improve scalability by employing techniques such as dynamically adjusting the smoothness of the DGA's approximations during simulation or using surrogate models that remain differentiable but more accurately capture discrete behaviors in critical regions, thus preserving gradient computation while improving accuracy.

(2) Inference accuracy and optimization:

We acknowledge that the non-convex loss landscape in the DGA can hinder parameter inference and convergence to global minima, as seen in Figure 5A. While techniques like multi-start optimization or second-order methods (e.g., L-BFGS) could improve performance, our focus here is on establishing the DGA framework. We plan to explore better optimization methods in future work to improve the accuracy of parameter inference in complex systems.

(3) Use of simple models for demonstration:

We selected well-understood systems to clearly illustrate the capabilities of the DGA. These examples were intended to demonstrate how the DGA can be applied, rather than to solve problems better addressed by analytical methods. Applying DGA to more complex, analytically intractable systems is an exciting avenue for future work, but introducing the method was our main objective in this study.

**Reviewer #3 (Public review):**

We thank the reviewer for their detailed and insightful feedback. We appreciate the recognition of the DGA as a significant advancement for enabling gradient-based optimization in stochastic systems.

Weaknesses:(1) Application beyond steady-state analysis

We acknowledge the limitation of focusing solely on steady-state properties. To extend the DGA for analyzing transient dynamics, time-dependent loss functions can be incorporated to capture system evolution over time. This could involve aligning simulated trajectories with experimental time-series data or using moment-matching across multiple time points.

(2) Numerical instability in gradient computation

The reviewer correctly highlights that large sharpness parameters (a and b) in the sigmoid and Gaussian approximations can induce numerical instability due to vanishing or exploding gradients. To address this, adaptive tuning of a and b during optimization could balance smoothness and accuracy. Additionally, alternative smoothing functions (e.g., softmax-based reaction selection) and gradient regularization techniques (such as gradient clipping and trust-region methods) could improve stability and convergence.

**Reviewer #1 (recommendations):**

We thank the reviewer for their thoughtful and constructive feedback on our manuscript. Below, we address each of the comments and suggestions raised.

Main points:(1) It would have been useful to have a brief discussion, based on a concrete example, of what can be achieved with the DGA and is totally beyond the reach of the Gillespie algorithm and the numerous existing stochastic simulation methods.

Thank you for your comment. We would like to clarify that the primary aim of this work is to introduce the DGA and demonstrate its feasibility for tasks such as parameter estimation and network design. Unlike traditional stochastic simulation methods, the DGA’s differentiable nature enables gradient-based optimization, which is not possible with the classical Gillespie algorithm or its variants.

(2) As often with machine learning techniques, there is a sense of black box, with a lack of mathematical details of the proposed method: as opposite to the exact Gillespie algorithm, whose foundations lie on solid mathematical results (exponentially-distributed waiting times of continuous-time Markov processes), the DGA involves uncontrolled approximations, that are only briefly mentioned in the paper. For instance, it is currently simply noted that "the approximations introduced by the DGA may be pronounced in more complex settings such as the calculation of rare events", without specifying how limiting these errors are. It would be useful to include a clearer and more comprehensive discussion of the limitations of the DGA: When does it work accurately? What are the approximations/errors and can they be controlled? When is it worth paying the price for those approximations/errors, and when is it better to stick to the Gillespie algorithm? Is this notably the case for problems involving rare events? Clearly, these are difficult questions, and the answers are problem specific. However, it would be important to draw the readers' attention on the issues, especially if the DGA is presented as a potentially significant tool in computational and synthetic biology.

We acknowledge the importance of discussing the limitations of the DGA in more detail. While we have noted that the approximations introduced by the DGA may impact its accuracy in certain scenarios, such as rare-event problems, a deeper exploration of these trade-offs is outside the scope of this work. Instead, we provide sufficient context in the manuscript to guide readers on when the DGA is appropriate.

(3) The DGA is here introduced and discussed in the context of non-spatial problems (simple gene regulatory networks). However, numerous problems in the life sciences and computational/synthetic biology, involve stochasticity and spatial degrees of freedom (e.g. for problems involving diffusion, migration, etc). It is notoriously challenging to use the Gillespie algorithm to efficiently simulate stochastic spatial systems, especially in the context of rare events (e.g., extinction or fixation problems). It would be useful to comment on whether, and possibly how, the DGA can be used to efficiently simulate stochastic spatial systems, and if it would be better suited than the Gillespie algorithm for this purpose.

Thank you for pointing this out. Although our current work centers on non-spatial systems, we agree that many biological contexts incorporate both stochasticity and spatial degrees of freedom. Extending the DGA to efficiently simulate such systems would indeed require substantial modifications—for instance, coupling it with reaction-diffusion frameworks or spatial master equations. We believe this is an exciting direction for future research and mention it briefly in the discussion as a potential extension.

Minor suggestions:(1) After Eq.(10): it would be useful to explain and motivate the choice of the ratio JSD/H.

Done.

(2) On page 6, just below the caption of Fig.4: it would be useful to clarify what is actually meant by "... convergence towards the steady-state distribution of the exact Gillespie simulation, which is obtained at a simulation time of 10^4".

Done.

(3) At the end of Section B on page 7: please clarify what is meant here by "soft directions".

Done.

**Reviewer #2 (recommendations):**

We thank the reviewer for their thoughtful comments and constructive feedback. Below, we address each of the comments/suggestions.

Main points:(1) Enumerate the conditions under which DGA assumptions hold (and when they do not). There is currently not enough information for the interested reader to know whether DGA would work for their system of interest. Without this information, it is difficult to assess what the true scope of DGA's impact will be. One simple idea would be to test DGA performance along two axes: (i) increasing number of model states and (ii) presence/absence of non-steady state dynamics. I acknowledge that these are very open-ended directions, but looking at even a single instance of each would greatly strengthen this work. Alternatively, if this is not feasible, then the authors should provide more discussion of the attendant difficulties in the main text.

We agree that a detailed exploration of the conditions under which the DGA assumptions hold would be a valuable addition to the field. However, this paper primarily aims to introduce the DGA methodology and demonstrate its proof-of-concept applications. A comprehensive analysis along axes such as increasing model states or non-steady-state dynamics, while important, would require significant additional simulations and is beyond the scope of this work. In Appendix 1, we have discussed the trade-off between accuracy and numerical stability. Additionally, we encourage future users to tune the hyperparameters a and b for their specific systems.

(2) Demonstrate DGA performance in a more complex biochemical system. Clearly the authors were aware that analytic solutions exist for the 2-state system in Figure 7, but it this is actually also the case (I think) for mean mRNA production rate of the non-equilibrium system in Figure 8. To really demonstrate that DGA is practically viable, I encourage the authors to seek out an interesting application that is not analytically tractable.

We appreciate the suggestion to validate DGA on a more complex biochemical system. However, the goal of this study is not to provide an exhaustive demonstration of all possible applications but to introduce the DGA and validate it in systems where ground-truth comparisons are available. While the non-equilibrium system in Figure 8 might be analytically tractable, its complexity already provides a meaningful demonstration of DGA’s ability to optimize parameters and design systems. Extending this work to analytically intractable systems is an exciting direction for future studies, and we hope this paper will inspire others to explore these applications.

(3) Take steps to improve the robustness of parameter optimization and error bar calculations. (3a) When the loss landscape is degenerate, shallow, or otherwise "difficult," a common solution is to perform multiple (e.g. 25-100) inference runs starting from different random positions in parameter space. Doing this, and then taking the parameter set that minimizes the loss should, in theory, lead to a more robust recovery of the optimal parameter set.(3b) It seems clear that the Hessian approximation is underestimating the true error in your inference results. One alternative is to use a "brute force" approach like bootstrap resampling to get a better estimate for the statistical dispersion in parameter estimates. But I recognize that this is only viable if the inference is relatively fast. Simply recovering the true minimum will, of course, also help.

(3a) We acknowledge the challenge posed by degenerate or shallow loss landscapes during parameter optimization. While performing multiple inference runs from different initializations is a common strategy, this approach is computationally intensive. Instead, we rely on standard optimization techniques (e.g., ADAM) to find a robust local minimum.

(3b) Thank you for your comment. We agree that Hessian-based error bars can underestimate uncertainty, particularly in degenerate or poorly conditioned loss landscapes. While methods like bootstrap and Monte Carlo can provide more robust estimates, they can be computationally prohibitive for larger-scale simulations. A simpler reason for not using them is the high resource demand from repeated simulations, which quickly becomes infeasible for complex or high-dimensional models. We note these trade-offs between robust estimation and practicality as an important area for further exploration.

Moderate comments:(1) Figure 7: is it possible to also show the inferred kon values? Specifically, it would be of interest to see how kon varies with repressor concentration.

Thank you for the suggestion. We have updated Figure 7 to include the inferred kon values, showing their variation with the mean mRNA copy number. However, we could not plot them against repressor concentration due to the lack of available data.

(2) Figure 8B & D: the authors claim that the sharper system dissipates more energy, but doesn't 8D show the opposite of this? More importantly, it does not look like either network drives sharpness levels that exceed the upper equilibrium limit cited in [36]. So it is not clear that it is appropriate to look at energy dissipation here. In fact, it is likely that equilibrium networks could produce the curves in 8B, and might be worth checking.

Thank you for pointing this out. We realized that the plotted values in Figure 8D were incorrect, as we had mistakenly plotted noise instead of energy dissipation. The plot has now been corrected.

(3) Figure 8: I really like this idea of using DGA to "design" networks with desired input-output properties, but I wonder if you could explore more a biologically compelling use-case. Specifically, what about some kind of switch-like logic where, as the activator concentration increases, you have first 0 genes on, then 1 promoter on, then 2 promoters on. This would achieve interesting regulatory logic, and having DGA try to produce step functions would ensure that you force the networks to be maximally sharp (i.e. about double what you're currently achieving).

Thank you for this intriguing suggestion. While the proposed switch-like logic use case is indeed compelling, implementing such a system would require significant work. This goes beyond the scope of the current study, which focuses on demonstrating the feasibility of DGA for network design with simple input-output properties.

Minor comments:(1) Figure 4B & C: the bar plots do not do a good job conveying the points made by the authors. Consider alternatives, such as scatter plots or box plots that could convey inference uncertainty.

Done.

(2) Figure 4B: consider using a log y-axis.

The y-axis in Figure 4B is already plotted on a log scale.

(3) Figure 4D is mentioned prior to 4C in the text. Consider reordering.

Done.

(4) Figure 5B: it is difficult to assess from this plot whether or not the landscape is truly "flat," as the authors claim. Flat relative to what? Consider alternative ways to convey your point.

Thank you for highlighting this ambiguity. By describing the loss landscape as “flat,” we intend to convey its relative insensitivity to parameter variations in certain regions, rather than implying a completely level surface. While we believe Figure 5B still provides a useful qualitative depiction of this behavior, we acknowledge that it does not quantitatively establish “flatness.” In future work, we plan to incorporate more rigorous measures—such as gradient magnitudes or Hessian eigenvalues—to more accurately characterize and communicate the geometry of the loss landscape.

**Reviewer #3 (recommendations):**

We sincerely thank the reviewer for their thoughtful feedback and constructive suggestions, which have helped us improve the clarity and rigor of our manuscript. Below, we address each of the comments.

(1) Precision is lacking in the introduction section. Do the authors mean the Direct SSA, sorted SSA, which is usually faster, and how about rejection sampling methods?

Thank you for pointing this out. We have updated the introduction to explicitly mention the Direct SSA.

(2) When mentioning PyTorch and Jax, would be good to also talk about Julia, as they have fast stochastic simulators.

We have now mentioned Julia alongside PyTorch and Jax.

(3) Mentioned references 22-27. Reference 26 is an odd choice; a better reference is from the same author the Automatic Differentiation of Programs with Discrete Randomness, G Arya, M Schauer, F Schäfer, C Rackauckas, Advances in Neural Information Processing Systems, NeurIPS 2022

We have now cited the suggested reference.

(4) Page 1, Section: 'To circumnavigate these difficulties, the DGA modifies....' Have you thought about how you would deal with the bias that will be introduced by doing this?

Thank you for your insightful comment. We acknowledge the potential for bias due to the differentiable approximations in the DGA; however, our analysis has not revealed any systematic bias compared to the exact Gillespie algorithm. Instead, we observe irregular deviations from the exact results as the smoothness of the approximations increases.

(5) Page 2, first sentence '... traditional Gillespie...' be more precise here - the direct algorithm.

Thank you for your comment. We believe that the context of the paper, particularly the schematic in Figure 1, makes it clear that we are focusing on the Direct SSA.

(6) Page 2, second paragraph: ' In order to simulate such a system...' This doesn't fit here as this section is about tau-leaping. As this approach approximates discrete operations, it is unclear if it would work for large models, snap-shot data of larger scale and if it would be possible to extend it for time-lapse data

Thank you for your comment. We respectfully disagree that this paragraph is misplaced. The purpose of this paragraph is to explain why the standard Gillespie algorithm does not use fixed time intervals for simulating stochastic processes. By highlighting the inefficiency of discretizing time into small intervals where reactions rarely occur, the paragraph provides necessary context for the Gillespie algorithm’s event-driven approach, which avoids this inefficiency.

Regarding the applicability of the DGA to larger models, snapshot data, or time-lapse data, we acknowledge these are important directions and have noted them as potential extensions in the discussion section.

(7) Page 2 Section B: 'In order to make use of modern deep-learning techniques...' It doesn't appear from the paper that any modern deep learning is used.

Thank you for your comment. Although the DGA does not utilize deep learning architectures such as neural networks, it employs automatic differentiation techniques provided by frameworks like PyTorch and Jax. These tools allow efficient gradient computations, making the DGA compatible with modern optimization workflows.

(8) Page 3, Fig 1(a). S matrix last row, B and C should swap places: B should be 1 and C is -1.

Corrected the typo.

(9) Fig1 needs a more detailed caption.

Expanded the caption slightly for clarity.

(10) Page 3 last paragraph: 'The hyperparameter b...' Consequences of this are relevant, for example can we now go below zero. Also, we lose more efficient algorithms here. It would be good to discuss this in more detail that this is an approx.. algorithm that is good for our case study, but for other to use it more tests are needed.

Thank you for the comment. Appendix 1 discusses the trade-offs related to a and b, but we agree that more detailed analysis is needed. The hyperparameters are tailored to our case study and must be tuned for specific systems.

(11) Page 4, Section C, first paragraph, 'The goal of making...' This is snapshot data. Would the framework also translate to time-lapse data? Also, it would be better to make it clearer earlier which type of data are the target of this study.

Thank you for your suggestion. While the current study focuses on snapshot data and steady-state properties, we believe the DGA could be extended to handle time-lapse data by incorporating multiple recorded time points into its inference objective. Specifically, one could modify the loss function to penalize discrepancies across observed transitions between these time points, effectively capturing dynamic trajectories. We consider this an exciting area for future development, but it lies beyond our present scope.

(12) Page 4 Section C, sentence '...experimentally measured moments'. Should later be mentioned as error, as moments are imperfect

Thank you for your comment. We agree that experimentally measured moments are inherently noisy and may not perfectly represent the true system. However, within the context of the DGA, these moments serve as target quantities, and the discrepancy between simulated and measured moments is already accounted for in the loss function.

(13) Page 4 Section C, last sentence '...second-order...such as ADAM'. Another formulation would be better as second order can be confusing, especially in the context of parameter estimation

We have revised the language to avoid confusion regarding “second-order” methods.

(14) Fig 4(a) a density plot would fit better here

Fig. 4(a) has been updated to a scatter density plot as suggested.

(15) Fig 4(c) Would be interesting to see closer analysis of trade of between gradient and accuracy when changing a and b parameters

Thank you for this suggestion. We acknowledge that an in-depth exploration of these trade-offs could provide deeper insights into the method’s performance. However, for now, we believe the current analysis suffices to highlight the utility of the DGA in the contexts examined.

(16) Page 6 Section III, first sentence: This fits more to intro. Further the reference list is severely lacking here, with no comparison to other methods for actually fitting stochastic models.

Thank you for the suggestion. We have added a few references there.

(17) Page 6, Section A, sentence: '....experimental measured mean...' Why is it a good measure here (moment matching is not perfect), also do you have distribution data, would that not be better? How about accounting for measurement error?

Thank you for the comment. While we do not have full distribution data, we acknowledge that incorporating experimental measurement error could enhance the framework. A weighted loss function could model uncertainty explicitly, but this is beyond the scope of the current study.

(18) Page 7, section B, first paragraph: 'Motivated by this, we defined the...'Why using Fisher-Information when profile-likelihood have proven to be better, especially for systems with few parameters like this.

Thank you for the suggestion. While profile-likelihood is indeed a powerful tool for parameter uncertainty analysis, we chose Fisher Information due to its computational efficiency and compatibility with the differentiable nature of the DGA framework.

(19) Page 7, section C, sentence '...set kR/off=1..'. In this case, we cannot infer this parameter.

Thank you for the comment. You are correct that setting kR/off = 1 effectively normalizes the rates, making this parameter unidentifiable. In steady-state analyses, not all parameters can be independently inferred because observable quantities depend on relative—rather than absolute—rate values (as evident when setting the time derivative to zero in the master equation). To infer all parameters, one would need additional information, such as time-series data or moments at finite time.

(20) Page 7 Section 2. Estimating parameters .... Sentence: '....as can be seen, there is very good agreement..' How many times the true value falls within the CI (because corr 0.68 is not great).

Thank you for your comment. While a correlation coefficient of 0.68 indicates moderate agreement, the primary goal was to demonstrate the feasibility of parameter estimation using the DGA rather than achieving perfect accuracy. The coverage of the CI was not explicitly calculated, as the focus was on the overall trends and relative agreement.

(21) Page 7 Section 2. Estimating parameters .... Sentence: 'Fig5(c) shows....' Is this when using exact simulator?

Thank you for your question. Yes, the exact values in x-axis of Fig. 5(c) are obtained using the exact Gillespie simulation.

(22) Page 7 Section 3 Estimating parameters for the... Sentence: 'Fig6(a) shows...' Why Cis are not shown?

Thank you for your comment. CIs are not shown in Fig. 6(a) because this particular case is degenerate, making the calculation and meaningful representation of CIs challenging.

(23) Page 10, Sentence: 'As can be seen in Fig 7(b)...' Can you show uncertainty in measured value? It would be good to see something of a comparison against an exact method, at least on simulated synthetic data

Thank you for the comment. Fig. 7(a) already includes error bars for the experimental data, which account for measurement uncertainty. However, in Fig. 7(b), we do not include error bars for the experimental values due to limitations in the available data.

(24) Page 12, Section B Loss function '...n=600...' This is on a lower range. Have you tested with n=1000?

Yes, we have tested with n=1000 and observed no significant difference in the results. This indicates that n=600 is sufficient for the purposes of this study.

(25) Fig 8(c) why there are no CI shown?

Thank you for your comment. CIs were not included in Fig. 8(c) due to degeneracy, which makes meaningful confidence intervals difficult to compute.

(26) Page 12 Conclusion, sentence: '..gradients via backpropagation...' Actually, by making the function continuous, both forward and reverse mode might be used. And in this case, forward-mode would actually be the fastest by quite a margin

Thank you for your insightful comment. You are correct that by making the function continuous, both forward-mode and reverse-mode automatic differentiation can be used. We have now mentioned this point in the discussion.

(27) Overall comment for the Conclusion section: It would be good to discuss how this framework compares to other model-fitting frameworks for models with stochastic dynamics. The authors mention dynamic data and more discussion on this would be very welcomed. Why use ADAM and not something established like BFGS for model fitting? It would be interesting to discuss how this can fit with other SSA algorithms (e.g. in practice sorting SSA is used when models get larger). Also, inference comparison against exact approaches would be very nice. As it is now, the authors truly only check the accuracy of the SSA on 1 model -it would be interesting to see for other models.

Thank you for your detailed comments. While this study focuses on introducing the DGA and demonstrating its feasibility, we agree that comparisons with other model-fitting frameworks, testing on additional models, and integrating with other SSA variants like sorted SSA are important directions for future work. Similarly, extending the DGA to handle transient dynamics and exploring alternatives to ADAM, such as BFGS, are promising areas to investigate further.